# Zebrafish macrophages convert physical wound signals into rapid vascular permeabilization

Zaza Gelashvili [1,2], Zhouyang Shen [1,2,3], Yanan Ma [1], Mark Jelcic[1,4] & Philipp Niethammer [1] ✉

Blood vessels near injury sites rapidly dilate, become permeable, and release serum and leukocytes into the wounded tissue to support healing and regeneration. How the vasculature senses distant homeostatic tissue perturbations within seconds-to-minutes remains incompletely understood. Using high-speed imaging of live zebrafish larvae, we monitor two hallmark vascular responses to injury: vessel dilation and serum exudation. By genetic, pharmacologic, and osmotic perturbation along with leukocyte depletion, we show that the $cPla_2$ nuclear membrane mechanotransduction pathway converts a ~ 50 µm/s osmotic wound signal into rapid vessel-permeabilization via perivascular macrophages, 5-lipoxygenase (Alox5a), and leukotriene A4 hydrolase (Lta4h). By revealing a previously undescribed physiological function of nuclear membrane mechanotransduction, we provide real-time insights into the long-range communication of wounds and blood vessels in intact tissue.

Realtime imaging of zebrafish larvae allows studying the spatio-temporal regulation of wound detection in vivo at timescales inaccessible to conventional biochemical or molecular biology methods[1,2]. Previous work in zebrafish, flies, and frogs, revealed extracellular nucleotides, bioactive lipids, $Ca^{2+}$ waves, reactive oxygen species (ROS), osmotic-, and ionic- gradients as physiological mediators of rapid wound detection[3–12].

We previously showed that zebrafish detect integumental breaches via the osmotic shock that occurs when environmental freshwater enters their tissues from the outside. Consequently, immersing zebrafish larvae in isotonic salt or sugar solutions suppresses neutrophil recruitment and acute breach closure by epithelial cells. Compared to other isotonic treatments, NaCl has a stronger inhibitory effect[4,5], which may be due to electric fields (EFs)[7], or other, yet unknown salt-sensing mechanisms. Osmotic shock at the wound induces local cell swelling and nuclear deformation that stretches the inner nuclear membrane (INM)[13]. Along with $Ca^{2+}$, INM tension ($T_{INM}$) recruits $cPla_2$ to the INM to release

arachidonic acid (AA). Lipoxygenases, cyclooxygenases, etc., further oxidize AA into bioactive lipids that control chemotaxis, cell differentiation, blood vessel tone and permeability, and many other processes[14–18]. Besides wound detection, the $cPla_2$ nuclear membrane mechanotransduction/shape sensing pathway has been reported to control confined cell migration, dendritic cell chemotaxis, mesenchymal stem cell differentiation, and the cell cycle[19–25]. At larval wounds and infection sites, downstream metabolites of $cPla_2$ rapidly recruit leukocytes[4,26,27]. At least in part, these leukocytes come from the blood circuit. To release them nearby wound sites, blood vessels can apparently sense wounds as fast or even faster than leukocytes. Although a plethora of vascular regulators have been described, how local tissue injury is detected and relayed to nearby vessels remains little studied in situ.

In this work, real-time intravital imaging reveals that perivascular macrophages mediate rapid vessel permeabilization at acute osmotic wounds through the $cPla_2$ nuclear membrane mechanotransduction pathway.

[1]Cell Biology Program, Memorial Sloan Kettering Cancer Center, New York, NY, USA. [2]Louis V. Gerstner, Jr. Graduate School of Biomedical Sciences, Memorial Sloan Kettering Cancer Center, New York, NY, USA. [3]Bloomberg-Kimmel Institute for Cancer Immunotherapy, Department of Oncology, Johns Hopkins University School of Medicine, Baltimore, MD, USA. [4]Fate Therapeutics, Inc, San Diego, CA, USA. ✉e-mail: niethamp@mskcc.org

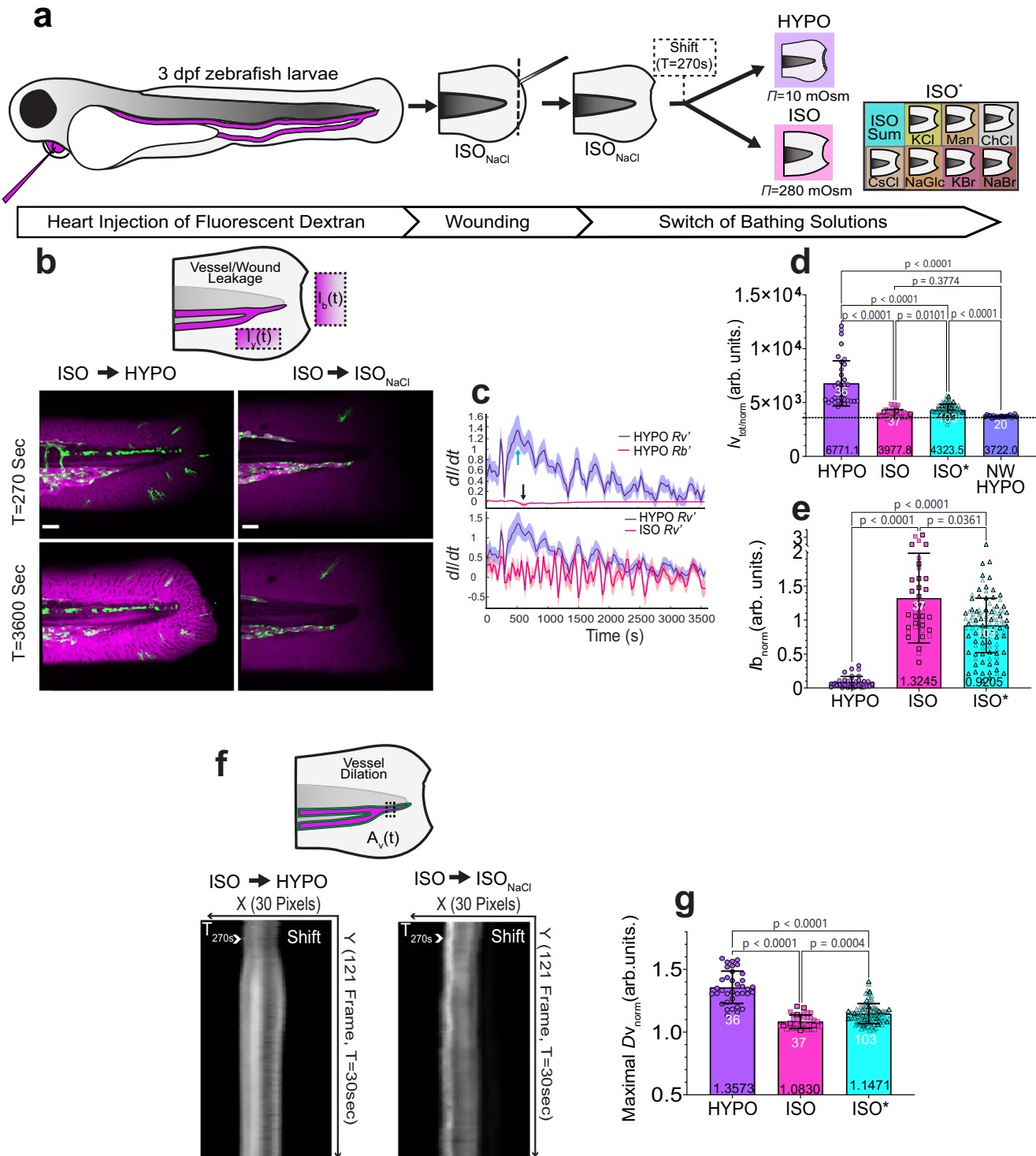

## Results

### Rapid blood vessel permeability is mediated by osmotic shock

To image rapid vessel responses to injury in vivo, we intracardially injected 2-3 days post fertilization (dpf) larvae with 70 kDa fluorescent dextran to label the circulating blood pool. Carefully avoiding damage to the vasculature, we amputated the tail fin tips of these larvae in E3 solution[28] adjusted to isotonicity with sodium chloride ($ISO_{NaCl}$) to roughly match the composition of vertebrate interstitial fluid (Fig. 1a). We previously showed that $ISO_{NaCl}$ prevents rapid leukocyte recruitment and epithelial wound closure in zebrafish larvae[4,5,29]. Wound detection can be triggered by transferring the injured larvae back to

fresh water such as E3 (HYPO). This ISO→HYPO switch of bathing solutions, temporally synchronizes the wound response in animals injured and imaged at different times, facilitating statistical integration. The individual contributions of osmotic pressure and ionic cues to wound detection can be distinguished by shifting into isotonic solutions of different salts or sugar compositions (ISO*).

Using spinning disk confocal microscopy, we followed vessel permeability by measuring the fluorescence intensity ($I\nu(t)$) of 70 kDa fluorescent dextran in a region of interest (ROI) adjacent to the caudal vein (Fig. 1b−e) after $ISO_{NaCl}$→HYPO, $ISO_{NaCl}$→$ISO_{NaCl}$, or $ISO_{NaCl}$→ISO* shifts. Besides permeabilization, we also measured

**Fig. 1 | Rapid blood vessel permeability is mediated by osmotic shock.**
**a** Schematic of the fluorescence microangiography workflow, tail fin wounding, and ionic/osmotic treatments. HYPO, regular E3 solution. $ISO_{NaCl}$, E3 adjusted to interstitial osmolarity with 135 mM NaCl. ISO*, E3 adjusted to isotonicity with other salts/osmolytes. Dotted line indicates amputation region, dotted box indicates time of solution switch (shift). **b** Cartoon scheme, depicting the regions of measurement for dextran permeability of vessels ($Iv$) and wounds ($Ib$). Left panel, confocal maximal intensity projection (MIP) of wounded 3dpf larvae before (t = 270 s) and after (t = 3600 s) switch of bathing solutions. Green, *kdrl*:eGFP fluorescence. Magenta, pseudo-colored 70 kDa dextran fluorescence. Scale bars, 50 μm. **c** Top panel, the rate ($dI\ dt^{-1}$) of vessel (purple) and wound (red) leakage during $ISO_{NaCl}$ to HYPO shift. Arrows indicate the maximal rate of change for vessel permeability (cyan) or wound permeability (black). Bottom panel, rate ($dI\ dt^{-1}$) of vessel leakage during $ISO_{NaCl}$ to HYPO (blue) or to $ISO_{NaCl}$ shifts. Lines, Average. Shaded error bars, SEM. Note, the blue curves represent the same, replotted

dataset. HYPO, n = 36 larvae; ISO, n = 37 larvae) (**d**) Quantification of normalized, integrated (between t = 0–3600 s) dextran leakage ($N_{tot/norm}$). note, NW= non-wounded. Dashed line is hypothetical no leakage baseline= 3600 (arb. units). **e** Quantification of normalized dextran leakage from the wound ($Ib_{norm}$) measured at t = 3600 s. **f** Cartoon scheme depicting region for vessel dilation ($Dv$). Left panel, kymographs of vessel diameter before and after switch of bathing solution. **g** Bar graph quantification of maximal, normalized endothelial diameter (max($Dv_{norm}$)), obtained from kymograph. Unless indicated, data were normalized by the mean of the first 10 frames (t = 0–270 s, i.e., preceding solution switching). Note, ISO* contains pooled isotonic salt/osmolyte treatments. (HYPO, n = 36 larvae; ISO, n = 37 larvae; ISO*, n = 103 larvae; NW HYPO, n = 20 larvae). Bar graph error bars, SD. White numbers, animals. Black numbers, mean of dataset. P values are indicated in figure and determined by unpaired, two-sided Kruskal–Wallis Test with Dunn's post-hoc test (**d, e, g**). Source data are provided in the Source Data file.

vessel diameter ($Dv(t)$) within a ROI comprising a vein/artery subsection (Fig. 1f, g).

Vessel permeability drastically rose upon hypotonic but not isotonic switching (Fig. 1c). Both, dextran leakage from the vessel into the interstitium, and its loss through the wound may influence perivascular dextran intensity. We measured dextran's wound leakage in a ROI next to the amputation site just outside the animal ($Ib(t)$) (Fig. 1b, e). Although wound leakage was very slow compared to vessel leakage ($Rv' \gg Rb'$) (Fig. 1c–e) presented a useful proxy for wound permeability/size.

To integrate the amount of dextran leaking from vessels over the course of the experiment, we summed up the respective, normalized $Iv$norm intensity values (i.e., determined the areas under the curves in Supplementary Fig. 1a). Since $Iv$norm is 1 when there is no wound-induced permeability change, a hypothetical $Iv$tot/norm value of 3600 A.U. indicates that no net-leakage of dextran has occurred over the 3600 s of the experiment. In all the $Iv$ bar-graphs, this theoretical baseline leakage is indicated by a dotted line. The measured baseline leakage in unwounded larvae ('NW-HYPO', Fig. 1d) is only slightly ( ~ 3%) higher than this theoretical minimum. Depending on which baseline is used for calculation, ISO/ISO* treatment blocks wound-induced vessel permeability by ~88/77%. All effect size approximations indicated below are based on the theoretical 3600 (arb. units.) $Iv$tot/norm baseline.

For comparing wound closure and vessel dilation as a function of osmolyte treatment, we plotted normalized wound leakage (i.e., $Ib$norm at 3600 s; Fig. 1e) and the maximal vessel dilation (i.e., max($Dv$norm(t)); Fig. 1g). Consistent with earlier observations[4,5], isotonic NaCl always showed the strongest inhibitory effect, very closely followed by the other non-ionic and ionic isotonicity treatments (Fig. 1d–e, 1g; Supplementary Fig. 1b–f; Supplementary Movie 1–3). Some of the differential osmolyte effects might reflect EF perturbation as previously proposed[7].

Overall, our data suggested that the rapid surge of vessel permeability was primarily due to osmotic shock. The response was not sensitive to zebrafish strain (AB vs. Casper) or size of the injected dextran (500 kDa vs. 70 kDa) (Supplementary Fig. 1g–m; Supplementary Movie 4, 5). So, we set out to dissecting its mechanism.

## Osmotic blood vessel permeabilization depends on Alox5a and Lta4h

The osmotic injury response of leukocytes is mediated by bioactive lipids of the eicosanoid pathway[4]. So, we wanted to test whether similar mechanisms control blood vessel permeabilization. To this end, we perturbed different branches of the eicosanoid cascade (Fig. 2a) through mutation and/or chemical antagonists: 12-lipoxygenase (ALOX12), 5-lipoxygenase (ALOX5), leukotriene A4 hydrolase (LTA4H), and cyclooxygenase[30]. ALOX12 (zebrafish

ortholog: Alox12) is enriched in skin and catalyzes the conversion of AA into 12(S)-HETE, Hepoxilins and other epithelial lipid mediators[31]. ALOX5 (zebrafish ortholog: Alox5a) is enriched in leukocytes but also expressed in non-myeloid tissues[4,32–35]. Its downstream metabolite 5-KETE attracts zebrafish leukocytes to tail fin wounds in response to osmotic and oxidative stress[4,6,26,27,36,37]. LTA4H (zebrafish ortholog: Lta4h) mediates leukotriene synthesis downstream of ALOX5. Published in situ hybridization data show *lta4h* is expressed in leukocytes[34,38]. Cyclooxygenases are abundantly expressed in immune and non-immune cells and generate AA-derived prostaglandins, which also contribute to cardiovascular regulation[39]. They are clinically targeted by Non-steroidal Anti-Inflammatory Drugs (NSAIDs), such as diclofenac, aspirin and others.

The mutation of *alox12* (*alox12*^mk218/mk218^; Fig. 2a; Supplementary Fig. 2a, 2g-h, 2k) did not alter vessel- or wound- permeability (Fig. 2b, c). By contrast, *alox5a*[36] or *lta4h* mutation (*alox5a*^mk211/mk211^, *lta4h*^mk219/mk219^; Supplementary Fig. 2i-j, 2l-m) suppressed dextran leakage from vesels after hypotonic shifting by ~55% or 52%, respectively (Fig. 2d–g; Supplementary Fig. 2c, 2e).

Pharmacological ALOX5 inhibition with licofelone (a cyclooxygenase/5-lipoxygenase inhibitor)[40] or MK886 (a FLAP inhibitor)[41], unlike inhibition with diclofenac[42], resembled these effects (Fig. 2h, i; Supplementary Fig. 3a–c). None of the ALOX5 perturbations altered wound permeability/closure (Fig. 2d–i; Supplementary Fig. 2d, 2f), or gross vessel morphology/area (Supplementary Fig. 3d).

Direct bath supplementation with the ALOX5 substrate AA partially restored vessel leakage under isotonic conditions without affecting wound permeability (Supplementary Fig. 3f-I, 3k-l). Adenosine triphosphate (ATP), which is released by osmotic shock and promotes rapid epithelial wound closure[5], strongly suppressed dextran leakage from wounds and modestly increased vessel leakage and dilation (Supplementary Fig. 3h-j, 3m). The ATP hydrolysis product adenosine (Ad) affected only vessel dilation, without altering vessel- or wound permeability.

Altogether, these experiments highlighted a key role for the Alox5a-Lta4h pathway in transducing osmotic vessel permeabilization. Given the pathway's well-established involvement in inflammatory signaling by myeloid cells[35,43–45], we wondered whether osmotic vessel permeabilization was mediated by immune cells.

## Blood vessel permeabilization via Alox5a-Lta4h requires macrophages

At the larval stage, zebrafish do not possess functional adaptive immunity yet[46]. Neutrophils and macrophages, the two main types of larval leukocytes, rapidly respond to tail fin injury, infections, and osmotic shock[1,4,26,27,37,47]. This response can be imaged in vivo using fluorescent markers expressed under the control of the leukocyte-specific *lyz* or *mpeg1.1* promoters, which light up neutrophils or macrophages, respectively (Supplementary Fig. 4a–b)[48,49]. Just as osmotic

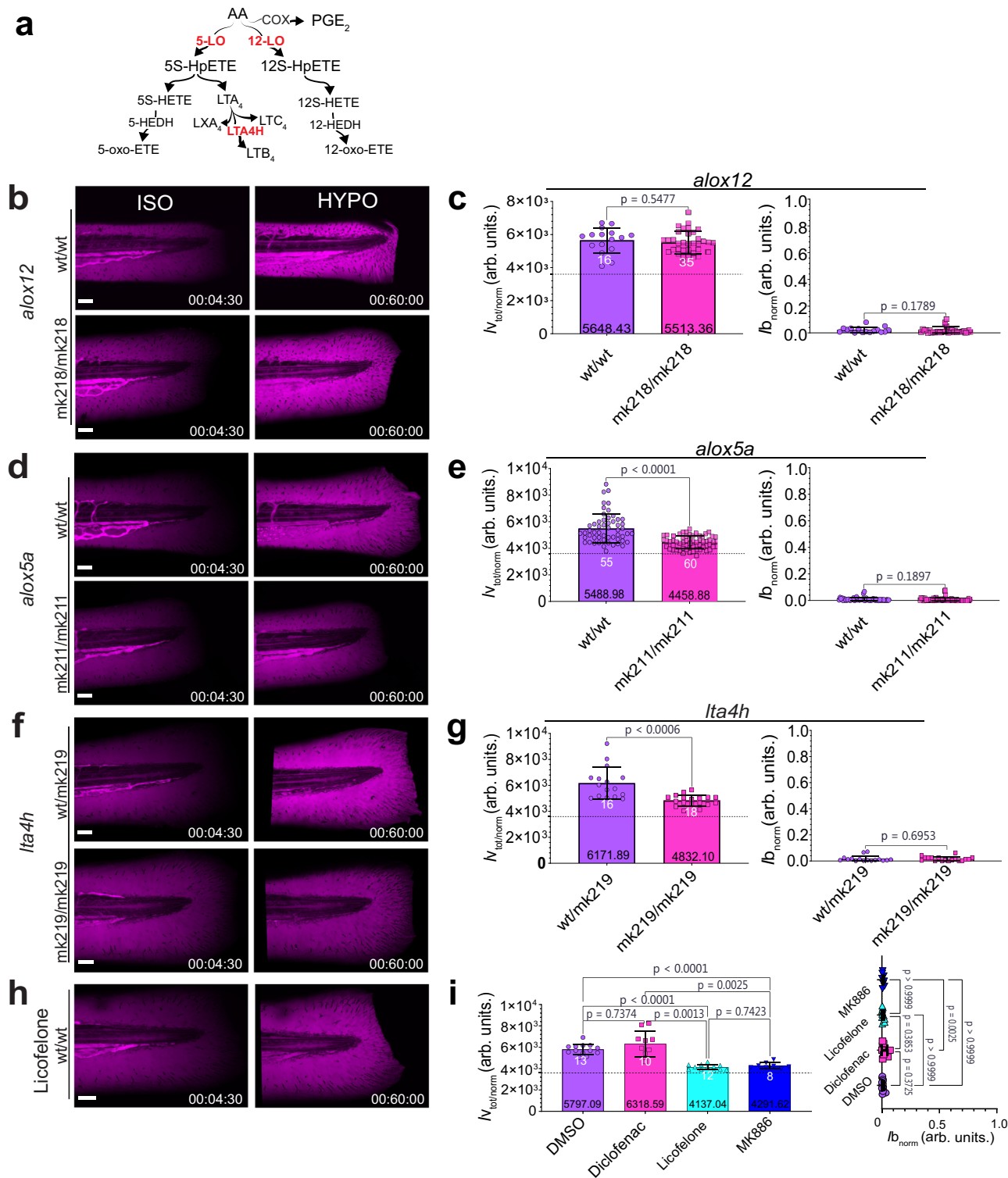

vessel permeabilization, osmotic leukocyte recruitment requires AA and Alox5a and is independent of Alox12[36]. Unlike osmotic vessel permeabilization, it initially does not depend on Lta4h activity[4], but instead on the 5-Hydroxyeicosanoid dehydrogenase (5-HEDH) product 5-KETE and its receptor OXER1 (zebrafish ortholog: Hcar1-4). The OXER1 pathway constitutes a little studied, branch of the ALOX5 pathway[37]. Underlining that osmotic leukocyte recruitment and vessel permeabilization involve partially distinct mechanisms, dextran

leakage from vessels was unaltered by *hcar1*-4 mutation (*hcar1*-4[mk214/mk214])[26] (Supplementary Fig. 4c).

We used the *lyz* or *mpeg1.1* promoters to specifically express nitroreductase (NTR) in neutrophils or macrophages. NTR converts the antibiotic metronidazole (Mtz) into a cytotoxic compound[50,51]. Ablating NTR-expressing leukocytes by bath supplementation of Mtz, allowed us to dissect neutrophil- and macrophage- contributions to osmotic vessel permeabilization (Fig. 3a-e; Supplementary Fig. 4d-i, 4j-

**Fig. 2 | Osmotic blood vessel permeabilization depends on Alox5a and Lta4h.**
**a** Simplified scheme of enzymatic derivatives of AA, the pathways enzymes tested in the study are highlighted in red. **b** Representative confocal maximum intensity projection (MIPs) images of magenta pseudo-colored 70 kDa dextran fluorescence in 3dpf wild-type animals (Casper) and (**b**) $alox12^{mk218/mk218}$ mutants before (t = 270 s, left panel) and after (t = 3600 s, right panel) $ISO_{NaCl}$ to HYPO shift. **c** Corresponding $alox12$ bar plot quantifications of integrated vessel (t = 0-3600 s, left plot), and wound leakage (t = 3600 s, right plot), respectively. P values are indicated in figure and determined by unpaired two-tailed t-tests with welch's correction or unpaired two-tailed Mann–Whitney U test, (wt/wt, n = 16 larvae; mk218/mk218, $n$ = 35 larvae). Representative confocal MIPs of (**d**) 3dpf wildtype and $alox5a^{mk211/mk211}$ mutants and corresponding bar plot quantifications (**e**) of vessel and wound leakage. P value and statistical significance was assessed using unpaired, two-tailed Mann–Whitney U test, (wt/wt, $n$ = 55 larvae; mk211/mk211, $n$ = 60 larvae). Representative confocal

MIPs of (**f**) 3dpf $lta4h^{wt/mk219}$ heterozygotes and $lta4h^{mk219/mk219}$ mutants and corresponding quantifications (**g**) and $p$ values were determined using unpaired, two-tailed Welch's $t$-tests or Mann–Whitney U tests, (wt/mk219, $n$ = 16 larvae; mk219/mk219, $n$ = 18 larvae). **h** Representative confocal MIPs of licofelone-treated 3dpf wildtype larva for vessel leakage after pretreatment with 50 μM Licofelone, 130 nM Diclofenac and 10 μM MK886, or DMSO (vehicle) and $ISO_{NaCl}$ to HYPO shifting with corresponding (**i**) integrated vessel leakage (left panel) and wound permeability (right panel) bar plot quantifications. P values are shown and were determined using unpaired, two-sided Welch's ANOVA with Dunnett's T3 post-hoc test or unpaired, two-sided Kruskal-Wallis Test with Dunn's post hoc test, (DMSO, $n$ = 13 larvae; Diclofenac, $n$ = 10 larvae; Licofelone, $n$ = 12 larvae; MK886, $n$ = 8 larvae). Dashed line is hypothetical no leakage baseline= 3600 (arb. units.). White numbers, animals. Black numbers, mean of dataset. Bar graph error bars, SD. Timestamp, hh: mm: ss. Scale bars, 50 μm. Source data are provided in the Source data file.

m). Mtz treatment of *lyz*-NTR2.0 larvae abolished the fluorescent *lyz*-marker signals and Sudan Black positivity in the tail fin, in line with complete neutrophil ablation (Fig. 3b, c; Supplementary Fig. 4j, 4k; Supplementary Movie 6). Yet, vessel leakage upon hypotonic shock remained unaltered (Fig. 3c; Figure S4d). In stark contrast, macrophage ablation reduced osmotic vessel permeability by ~54%, i.e., akin to Alox5a-Lta4h pathway perturbation (Fig. 3d, e; Supplementary Fig. 4f; Supplementary Movie 7). If vessel permeabilization via this pathway depends on macrophages, macrophage-deficient animals should be insensitive to ALOX5 inhibition, as observed (Fig. 3f, g; Supplementary Fig. 4h). None of the treatments affected rapid wound permeability/closure (Fig. 3b–g; Supplementary Fig. 4e, 4g, 4i), and macrophage ablation also did not alter early neutrophil recruitment to the wound (Supplementary Fig. 4l, 4m).

Part of the AA that drives Alox5a-dependent leukocyte migration is generated by $cPla_2$ that adsorbs to the stretched INM of osmotically swollen cells[4,6,52–54]. Was $cPla_2$ also fueling osmotic vessel permeabilization? More specifically, does nuclear membrane mechanotransduction via this enzyme allow macrophages to sense distant wounds in situ?

### $cPla_2$ mediates osmotic vessel permeabilization and wound sensing by macrophages

$cPLA_2$ drives LTA4H-dependent eicosanoid production in peritoneal mouse macrophages[55], Human[56] (e.g., lung) macrophages show nuclear $cPLA_2$, ALOX5 and LTA4H expression. Together with zebrafish single cell mRNA profiling data[32,35,57–59], this underscores that the pathway is conserved in macrophages across phylae. The $cPla_2$ mutants ($pla2g4aa^{mk220/mk220}$; Supplementary Fig. 5a-e) showed a ~59% reduction in osmotic vessel leakage with unaltered wound permeability (Fig. 4a, b).

To test whether and where macrophages osmotically activate $cPla_2$ upon larval injury, we expressed fluorescently tagged $cPla_2$ ($cPla_2$-mKate2)[4,6,13,20,21,52] using the *mpeg1.1* promoter (Fig. 4c–f; Supplementary Fig. 5f). The animals were bathed in normal hypotonic (i.e., fresh water) or isotonic solution during the laser-wounding procedure. Under hypotonic but not isotonic bathing conditions, tail fin wounding triggered a pulse of $cPla_2$-mKate2-INM adsorption, which passed through the field of view within ~10 s (Fig. 4g; Supplementary Fig. 5f; Supplementary Movie 8). Macrophages neighboring the wound (~20 μm) showed constitutive $cPla_2$ binding to the INM. By contrast, macrophages at the vasculature (i.e., "perivascular") showed highly reversible binding. Furthermore, none of the genetic $cPla_2$-Alox5a-Lta4h larvae mutants altered gross vessel morphology/area, or baseline vessel permeability in non-wounded mutants compared to *wt* siblings (Supplementary Fig. 6a-c).

These observations argue that macrophages nearby the vasculature directly detect distant wounds via osmotic $cPla_2$ activation. Two-photon resonance scanning microscopy corroborated these rapid dynamics at higher time-resolution (Fig. S7; Supplementary Movie 9). The respective translocation-wave propagates at ~50 μm/s through the tissue, i.e., consistent with interstitial sodium or chloride diffusion.

Remarkably, endothelial cells at comparable wound distances did not show similar $cPla_2$ responses (Supplementary Fig. S8a–c; Supplementary Movie 10), suggesting that the nuclei of perivascular macrophages are particularly sensitive to mechanical/osmotic stress (Fig. 4h).

## Discussion

Altogether, our data identify perivascular macrophages as, physico-chemical sentinels of osmotic tissue homeostasis and epithelial barrier integrity. This study focuses on osmotic signaling, which for zebrafish is obviously crucial, as their entire body surface is exposed, and their epidermal defence systems adapted to detect hypotonic fresh water influx. However, orthogonal immune adaptations to specific luminal fluid environments might also exist in land-living vertebrates. A hypotonic saliva environment is maintained in the upper digestive tract of humans[60]. Its disruption causes healing and immune defects, for instance, in Sjogren's symptom[61]. Whether saliva mediates osmotic surveillance of oral or oesophageal barrier linings remains to be seen. Notably, the relevance of our findings may transcend osmotic shock: During Ventilation Induced Lung Injury (VILI), mammalian lung macrophages detect physical tissue stretch caused by air inflation as signal to augment inflammation[62,63]. Although VILI does not involve osmotic shock as physical trigger, similar mechanotransduction mechanisms might be at play: VILI is known to involve both, activation of the ALOX5-LTA4H pathway[64] and macrophages[65,66], just like osmotic vessel permeabilization in fish.

Nuclear mechanotransduction is still mostly studied in cultured cells and reconstituted systems[67,68]. Work directly addressing its physiological roles in vivo remains scarce. Thus, our study bridges a gap between cell biology and animal physiology via a quantitative, intravital imaging approach paired with genetics. Our work leaves some finer details open. For example, our data remain neutral on whether other cell types contribute to wound-induced vessel permeabilization besides macrophages (Supplementary Fig. 9). AA or other eicosanoids generated by fibroblasts or epithelial cells at the wound margin[4] may reach perivascular macrophages through transcellular diffusion[69]. We estimate that osmotic signals and the macrophage-ALOX5 axis account for ~90% and ~60% of rapid, wound-induced vessel permeabilization, respectively. Hence, other mechanisms likely contribute.

Our denotation "perivascular" is not supposed to refer to a specific type of macrophage, only to their position in the tissue. It is intriguing to speculate but beyond the scope of our study that macrophages near blood vessels are genetically or metabolically poised to sense biomechanical cues. Previous data suggest that nuclear membrane mechanotransduction through $cPla_2$ is facilitated by F-actin and lamin A/C perturbation[6]. Possibly, macrophages with softer, less constrained nuclei are more responsive to physical perturbation. Nuclear mechanics may be subject to differentiation/polarization state or microenvironmental signals, e.g., from vessels.

Suppression of macrophage-mediated vessel permeabilization does not alter rapid neutrophil recruitment. Yet, our work does not

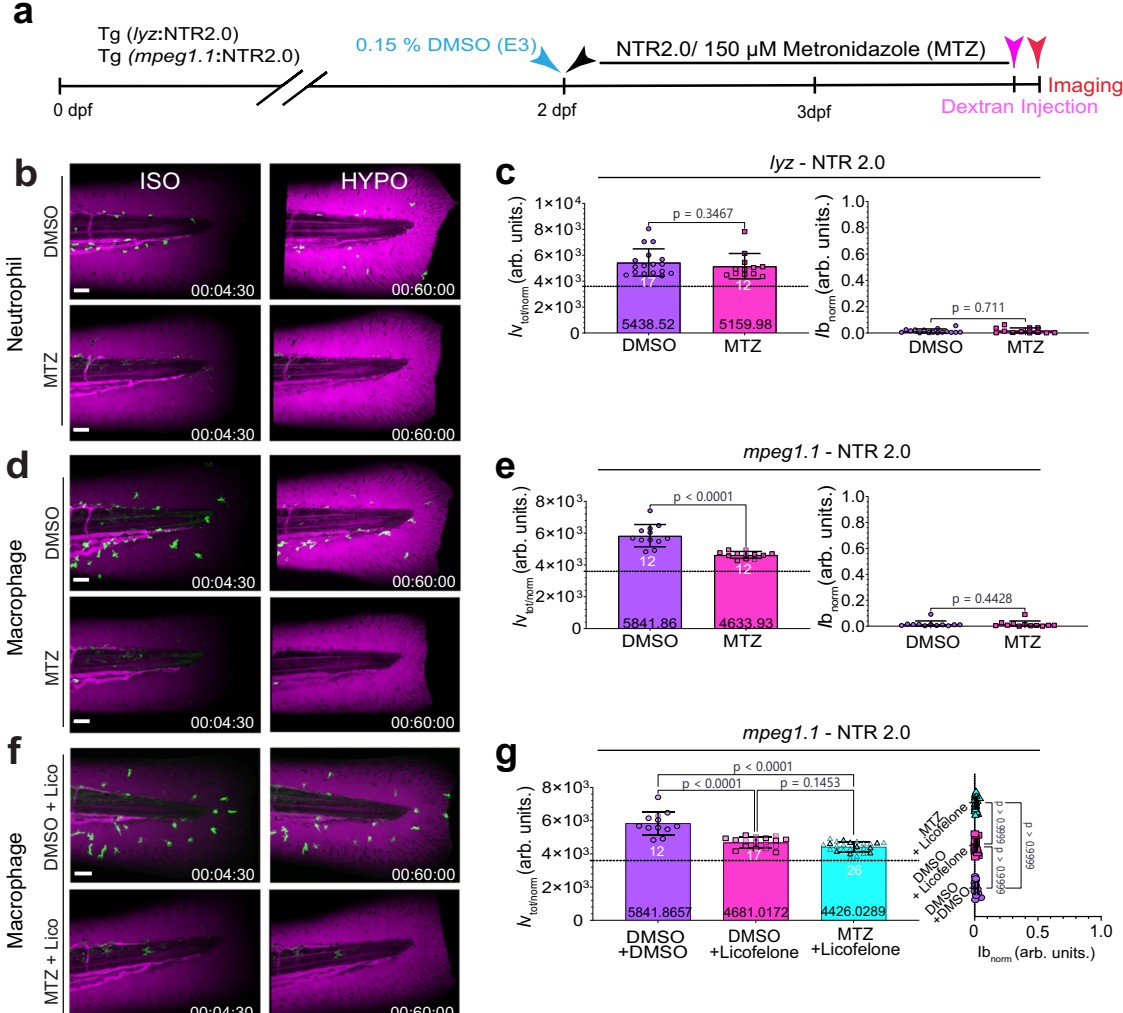

**Fig. 3 | Blood vessel permeabilization via Alox5a-Lta4h requires macrophages.**
**a** Experimental timeline for metronidazole-induced depletion of macrophages or neutrophils in 3 dpf Tg(*mpeg1.1*: YFP-NTR2.0) or Tg(*lyz*: YFP-NTR2.0) larvae, respectively. Blue arrow, start of the vehicle treatment (0.15% DMSO). Black arrow, start of the metronidazole (150 μM MTZ) treatment. Magenta arrow, dextran injection. Red arrow, start of imaging. **b** Representative confocal maximum intensity projections (MIPs) of pseudo-colored 70 kDa dextran fluorescence in 4dpf Tg (*lyz*:NTR2.0) larvae treated with DMSO or 150 μM MTZ before (t = 270 s; left panel) and after (t = 3600 s; right panel) ISO$_{NaCl}$ to HYPO shift with corresponding (**c**) *lyz*-NTR2.0 bar plot quantifications of integrated vessel leakage (t = 0–3600 s; left plot) and normalized wound permeability (t = 3600 s; right plot). *P* value is shown and calculated using an unpaired, two-tailed Mann–Whitney *U* test, (DMSO, n = 17 larvae; MTZ, n = 12 larvae). **d** Representative confocal MIPs of magenta pseudo-colored 70 kDa dextran fluorescence in 4dpf Tg(*mpeg1.1*:NTR2.0) larvae and 150 μM MTZ depletion with (**e**) *mpeg1.1*-NTR2.0 corresponding vessel and wound

leakage bar plot quantifications. P values were determined using unpaired, two-tailed Welch's t-tests or unpaired, two-tailed Mann–Whitney U tests, as appropriate, (DMSO, n = 12 larvae; MTZ, n = 12 larvae). **f** Representative confocal MIPs of pseudo-colored 70 kDa dextran fluorescence in 4dpf Tg(*mpeg1.1*:NTR2.0) larvae pretreated with Licofelone at indicated conditions with corresponding (**g**) *mpeg1.1*-NTR2.0 normalized vessel leakage (t = 0-3600 s; left plot) and normalized wound permeability (t = 3600 s; right plot) bar plot quantifications. Statistical significance was assessed using unpaired, two-sided Welch's one-way ANOVA with Dunnett's T3 multiple comparisons or unpaired, two-sided Kruskal–Wallis tests with Dunn's post hoc comparisons, as appropriate, (DMSO, n = 12 larvae; Licofelone, n = 17 larvae; Licofelone + MTZ, n = 26). Green, pseudo-colored *lyz or mpeg1.1*: YFP-NTR2.0, fluorescence. Magenta, pseudo-colored 70 kDa dextran fluorescence. Dashed line is hypothetical no leakage baseline= 3600 (arb. units.). White numbers, animals. Black numbers, mean of dataset. Bar graph error bars, SD. Timestamp, hh:mm:ss. Scale bars, 50 μm. Source data are provided in the Source data file.

address the long-term effects of abrogating immediate vessel permeabilization. The exudated serum may, for instance, provide an anabolic "jump-start" for tissue restoration. Regenerative and tumor promoting roles of macrophages are widely recognized[70–72]. Serving as a mechanically controlled "emergency-valve" is in line with these functions. The role of nuclear membrane mechanotransduction in perivascular macrophage biology[73] deserves further attention.

## Methods
### Zebrafish Husbandry
Adult wild-type and mutant Casper[74] Zebrafish (Danio rerio) strains and larvae were maintained[28] and subjected to experiments according

to the institutional compliance and approval of the animal ethics committee, the Institutional Animal Care and Use Committee (IACUC) and the Research Animal Resource Center (RARC) of the Memorial Sloan Kettering Cancer Center (MSKCC) (protocol no. 11-01-002). The adult fish are reared in either 2.8 or 6 L polycarbonate tanks at animal density 8 fish L$^{-1}$, with a photoperiod of 14:10 light:dark cycles and maintained in salinity-conditioned system water surveillance, (pH = 7.53) sodium bicarbonate (Proline, SC12A), conductivity 700-850 μS cm$^{-1}$ (Instant Ocean Sea Salt (#SS1-160P)) with TGP = 99.4% (Total Gas Pressure) and DO = 8.18 ppm (Dissolved Oxygen=8.155 mg L$^{-1}$) at 28 °C. The zebrafish are fed twice a day with live feed consisting of rotifers or artemia, followed by processed dry spirulina pellets (Zeigler). All

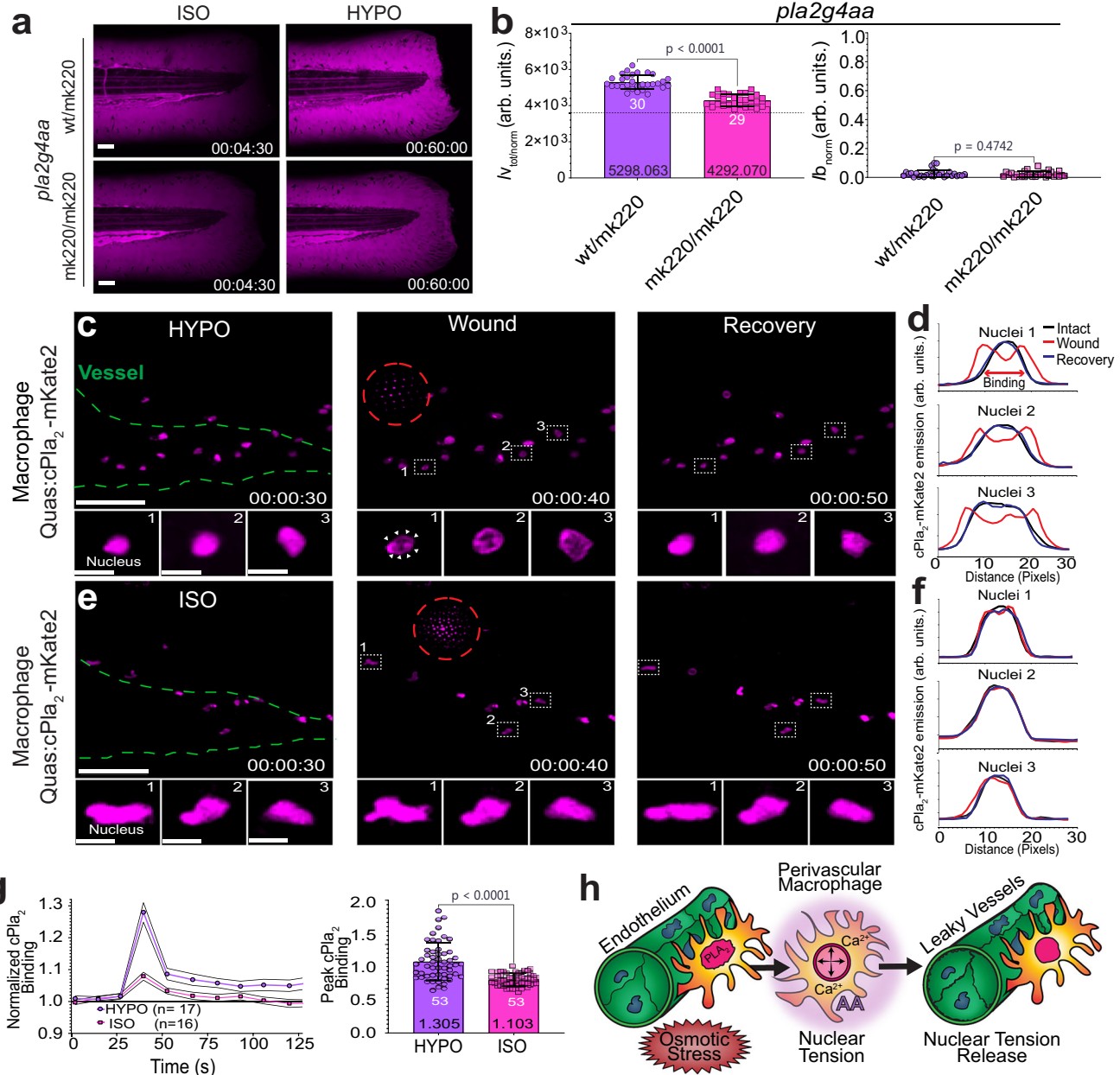

**Fig. 4 | cPla₂ mediates osmotic vessel permeabilization and wound sensing by macrophages. a** Left panel, representative confocal maximum intensity projection (MIPs) images of pseudo-colored 70 kDa Dextran before (t = 270 s) and after (t = 3600 s) ISO$_{NaCl}$ to HYPO shift in *pla2g4aa*$^{wt/mk220}$ and *pla2g4aa*$^{mk220/mk220}$ mutants. Magenta, pseudo-colored 70 kDa dextran fluorescence. Time stamp, hh:mm:ss. Scale bars, 50 µm. **b** Quantification of vessel (left bar plot) and wound leakage (right bar plot) in homozygous (*pla2g4aa*$^{mk220/mk220}$) and heterozygous (*pla2g4aa*$^{wt/mk220}$) cPla₂ mutant animals. P values are indicated and determined by unpaired, two-tailed Welch's t-test (left panel) or unpaired, two-tailed Mann–Whitney U test (right panel), (wt/mk220, n = 30 larvae; mk220/mk220, n = 29 larvae). Dashed line, hypothetical no leakage baseline (3600 arb. units.). **c** Representative MIPs of 3dpf Tg(*mpeg1.1*:cPla₂-mKate2) larvae before and after laser injury. Magenta, pseudo-colored cPla₂-mKate2 emission. Center of UV-blast (dashed, red circle), green dotted line (vessels). Note, white arrowheads indicate nuclear membrane adsorption and corresponding (**d**) line profiles of cPla₂-mKate2 fluorescence. **e** Representative confocal MIPs of Tg(*mpeg1.1*: cPla₂-mKate2) 3dpf in ISO$_{NaCl}$ bathing solution and corresponding (**f**) line profiles of cPla₂-mKate2 fluorescence in the ROI-labeled cells. Time stamp, hh:mm:ss. Scale bars, 50 µm. Inset scale bars, 10 µm. **g** Left panel, plot of cPla₂-mKate2-INM binding dynamics. UV laser injury is induced at t = 40 s. INM-binding of cPla₂-mKate2 is quantified as a ratio of perinuclear to nucleoplasmic and normalized to its initial (t = 0 s) fluorescence emission value. Lines, Average. Error Margins, 95% CI. Right panel, bar plot showing comparison of peak translocation. Note, the panel depicts two different representations of the same dataset. P value is indicated and calculated using unpaired, two-tailed Mann-Whitney U test, (HYPO, n = 17 larvae, N = 53 nuclei; ISO, n = 16 larvae, N = 53 nuclei). **h** Simplified cartoon scheme. Perivascular macrophage nuclei are reversibly stretched by osmotic wound signals. In the presence of Ca²⁺, nuclear membrane tension causes arachidonic acid (AA) release by cPla₂. AA is metabolized into a vessel permeabilizing lipid mediator. White numbers, (**a**) number of larvae, (**g**) number of analysed nuclei. Black numbers, mean of dataset. bar plot error bars, SD. Source data are provided in the Source data file.

anesthesia was conducted with 0.2 mg ml⁻¹ 3-amino benzoic acid ethyl ester (Sigma, MS-222, E10521), (pH=7.0), buffered in 0.5 mg ml⁻¹ anhydrous sodium phosphate dibasic (Fisher, BP332-500). The embryos were staged by dpf (days-post fertilization). Sex was

indeterminate at 2.5–4 dpf and all experiments supporting findings of the manuscript were conducted at these larval stages. The animal embryos were collected from natural spawning and raised in standard hypotonic E3 containing 0.1% (w/v) methylene blue (Sigma-Aldrich,

M9140) for first 24hrs followed by E3 medium (5 mM NaCl (Sigma-Aldrich, S7653), 0.17 mM KCl (Sigma-Aldrich, P9333), 0.33 mM CaCl$_2$ (Sigma-Aldrich, C5670), 0.33 mM MgSO$_4$ (Sigma-Aldrich, M7506)) in 100 mm petri dishes(Fisher Scientific, FB0875713).

## Transgenesis, plasmid construction and in vitro transcription

Fertilized Casper zebrafish embryos were collected and injected at the one-cell stage into the cytoplasm with a laser-pulled borosilicate glass microneedle (H = 370, FIL = 5, VEL = 50, DEL = 245, PUL = 140; P-2000 Sutter Instrument CO.) and a Nanoject II™ microinjector (Drummond Scientific, Broomball, PA)[75]. Plasmids were assembled using Gateway multisite cloning kit using LR Clonase™ (Invitrogen; C12537-023) and multicloning sites between the gateway att sites flanking cassette of p5E, pME and p3E plasmids that were recombined into Tol2kit[76] destination vectors pDESTTol2CG2* or pDEST-crybb1 for generating following Q-system animals[77]: Tg(kdrl:Qf2; p5E-kdrl Addgene Catalog# 78687)[78]; Tg(mpeg1.1:Qf2; p5E-mpeg1.1, Addgene Catalog# 75023)[79,80]; Tg(lyz:Qf2; p5E-LyzC Addgene Catalog# 135200)[79] or Tg(Quas:eGFP; pME-eGFP Tol2kit v1.2-383)[76,80]; Tg(5xQuas:GAP-tagYFP-P2A-NfsB_Vv)[50] and Tg(Quas:cPla$_2$-mKate2-P2A-eGFP-KDEL) or transgenic Casper zebrafish. The latter plasmid was constructed from Cytosolic Phospholipase A2 (cPla$_2$, Ensembl: ENSDARG00000024546)[1] where the open reading frame was amplified by PCR from a zebrafish cDNA clone (Open Biosystems, 9037889) using custom designed primers bearing unique palindromic overhangs, Bsu15I and EcoRI:(Fwd:5'-AGTCAtcgatGCCA-CAATGTCCAACATTATAG3'; (Rev:5'-CAGTGTGGCT GTGGGAGCTGGAGgaattcGACT-3'). The PCR product was RE digested, gel-purified(Qiagen, 28106) ligated into pDONOR221 containing pME gateway compatible att sites and subsequently fused in-frame to mKate2 (Evrogen) as a C-terminus fluorescent tag with a 15 amino-acid GS-enriched linker[6]. The p3E entry vector was fused in-frame with self-cleaving P2A peptide in N-terminus of eGFP followed by endoplasmic reticulum localization signal KDEL and SV40 poly(A). The final transgenesis constructs were mixed and ~2.7 nL of 25 ng μL$^{-1}$ of each plasmid co-injected with 25 ng μL$^{-1}$ Tol2kit transposase mRNA, transcribed from NotI linearized pCS2FA-transposase plasmid with mMESSAGE mMACHINE SP6 reverse Transcription kit (Thermo Scientific, AM1340) in wt or Casper Larvae. Among the injected larvae, fluorescence-positive siblings in the heart (cmlc2:eGFP)[76], lense (crybb1:mKate2)[81] or chorion (he1:eCFP)[50] were selected and raised in husbandry and backcrossed at sexual maturity to Casper fish. For establishing stable and constitutive tissue-specific fluorescent transgenic zebrafish lines, Qf2 founders were crossed with Quas animals and their progeny were identified through both transgenesis marker and indicated fluorescence in the designated tissue or cell type[77].

## Reagents, Isotonic reconstitution, and eicosanoid pathway Inhibition

For osmotic surveillance, a battery of iso-osmotic media was constituted by supplementing standard hypotonic E3 media ($\Pi$(osmolality)=10 mOsm) with the following osmolytes: 135 mM Na$^+$·Cl$^-$ (Sigma-Aldrich, S7653); K$^+$·Cl$^-$ (Sigma-Aldrich, P9333); K$^+$·Br$^-$ (Sigma-Aldrich, P9881); Na$^+$·Br$^-$ (Sigma-Aldrich, 220345); Cs$^+$·Cl$^-$ (Sigma-Aldrich, C3011); Na$^+$·C$_6$H$_{11}$O$_7^-$ (TCI Chemicals, G0041); C$_5$H$_{14}$NO$^+$·Cl$^-$ (Fisher Scientific, 50213240) and 270 mM D-mannitol (Fisher Scientific, BP686), culminating in ISO$_{NaCl}$ or ISO* ($\Pi$=280 mOsm) E3 media with the indicated reagents[4]. The final osmolality of solutions was approximated with ddH$_2$O (18.2MΩ·cm, Elga PURELAB®) calibrated handheld optical salinity refractometer (RHS-10ATC Cole-Parmer, EW-81150-31) - 1.0050:1.025-1.030 $d_{20}^{20}$SGSalt$^{-1}$ (-8:35-40 ‰ PPT) for hypotonic ($\Pi$=10 mOsm) and all isotonic ($\Pi$=280 mOsm) E3 reagents respectively and used for Imaging and analysis. For reconstitution experiments the nucleotides, their derivatives were conducted by ISO$_{NaCl}$ to ISO$_{NaCl}$

shifting approach, where the latter was constituted with ISO$_{NaCl}$ alone as vehicle or supplemented with 5 mM following nucleotide biomolecules: adenosine 5'-triphosphate disodium salt hydrate (ATP; Sigma-Aldrich, A26209) and adenosine (Ad; Sigma-Aldrich, A9251)[5]. The polyunsaturated arachidonic acid (20:4(5,8,11,14-all-cis-eicosatetranoic acid); Sigma-Aldrich, A3611) was administered in parallel to anesthetized and wounded 3dpf Tg (kdrl:Qf2:Quas:eGFP) larvae at a final concentration of 5 μM in ISO$_{NaCl}$ at 5 min during the imaging experiment. For eicosanoid pathway pharmacological inhibition experiments, Tg (kdrl:Qf2:Quas:eGFP) larvae were pre-incubated for 30-45 or 180 min in hypotonic ($\Pi$=10 mOsm) E3 supplemented with the following compounds: 50 μM Licofelone (Cayman Chemical Company, 100007692), 10 μM MK886 (Cayman Chemical Company, 10133) or 130 nM Diclofenac (Sigma-Aldrich, 1188800), respectively[36]. The imaging medium also contained the same concentration of the indicated inhibitors dissolved in hypotonic E3 ($\Pi$=10 mOsm) and administered 5 min after wounding. All water-insoluble biomolecules and inhibitors were dissolved at a maximal concentration of 1% dimethyl sulfoxide, DMSO (Millipore Sigma, 276855) hypotonic ($\Pi$=10 mOsm) or ISO$_{NaCl}$($\Pi$=280 mOsm) E3 as a vehicle control.

## Larvae preparation, microangiography, and tailfin wounding

Zebrafish larvae (2.5–4 dpf) were screened for transgenes, anesthetized, and oriented with their left lateral side facing up. They were positioned within slots imprinted from a custom-designed mold cast (NIH 3D Print Exchange, ID: 3DPX-021299) in 2% agarose (Fisher Scientific, BP160) dissolved in hypotonic ($\Pi$ = 10 mOsm) E3 medium. For microangiography, approximately 5 nL of 70 kDa dextran-tetramethylrhodamine (Fisher Scientific, D1818) or 500 kDa FITC-dextran (Fisher Scientific, D7136) dissolved in 1× phosphate-buffered saline (PBS; Sigma-Aldrich, 79382) at 1 mg/mL was microinjected into the Common Cardinal Vein (CCV) or the larval atrium, between the gill and pectoral fin. The injection was performed using a borosilicate microneedle (H = 410, FIL = 5, VEL = 40, DEL = 245, PUL = 160; P-2000 Sutter Instrument Co.) and a microinjector (Drummond Scientific, Broomball, PA) under the guidance of a stereomicroscope (Discovery V.8, Zeiss).

Larvae with successful perfusion of fluorescent dextran dyes in the caudal arteries and veins were identified using a coaxial fluorescence dissection stereomicroscope (MVX10, Olympus) equipped with an MVPLAPO 1X objective (NA 0.25, WD 65 mm, FN22) and a mercury lamp (LM200B1-A, Prior Scientific) with dichroic mirror sets (MVX-RFA; 540/35 U-MGFPHQ/XL, 625/55 U-MRFPHQ/XL). For microangiography in non-casper background, dextran was injected in anesthesized AB larvae that were maintained in 0.2mM N-phenylthiourea (PTU, Sigma-Aldrich, P7629) E3 to prevent pigment formation[3]. The injection site was accessed by making a small incision in the region between the gills and beneath the pectoral fin of 2.5-4 dpf larvae. Injections were performed percutaneously using a fine borosilicate microneedle (tip diameter ~ 10 μm), which gently penetrated the epidermis to reach the circulation. Successful entry into the circulation was confirmed by observing a slow retrograde flow column of red blood cells into the microneedle. Care was taken to avoid damage caused by dextran ejection; larvae with such damage were excluded from further experiments. For the 500 kDa dextran, working solution is more viscous, the injection needle was slowly retracted from the circulation after ejection, rather than being removed directly, to prevent dye leakage (evisceration).

For tailfin imaging experiments, anesthetized larva were flat-mounted on their right side in a 35 mm glass-bottom microscopy imaging dish (MatTek Corporation, P12G-1.5-14 F) and immersed in ~200 μL of 0.8% (w/v) low-melting agarose (Goldbio, A-204-100) dissolved in ISO$_{NaCl}$ ($\Pi$=280 mOsm) E3 medium. For tailfin wounding, multiple larva were aligned in parallel, and the tips of the larval caudal fins were amputated at the posterior end of the notochord using a

tungsten microblade needle (Fine Science Tools, 10318-14). Care was taken to avoid damaging the notochord or caudal vessels during the solidification of the low-melting agarose[4,26]. For non-wounded imaging experiments, the anesthesized zebrafish embryos were reared in borosilicate dish (Corning, 3160-60) and handled with manual pipette controller attached (Sigma-Aldrich, P7924) to borosillicate pasteur pipette (Corning, #7800;7095D-9) to prevent any damage to the end of tailfin from larva handling during dextran injection and immobilization. After immobilization, the embedded live larvae were enveloped in 200 µL of ISO$_{NaCl}$ ($\Pi$=280 mOsm) containing tricaine, to prevent desiccation. The immobilized larvae are allowed to rest at 28 °C for approximately 15 min to restore cardiac function in the trunk. Following this, the tailfin region of the embedded larvae was carefully freed from the excess low-melting agarose using a tapered micro spatula (Fine Science Tools, 10089-11) and transferred to a pre-heated stage-top incubator (INU-TIZ-D35).

## Spinning disk confocal microscopy

Tailfin amputation and non-wounded Imaging experiments were completed at 28 °C with the heated imaging chamber (INUG2 KIW, TOKAI-HIT) and sub-stage heater (AirTherm ATX In vivo Scientific, WPI inc.), in an inverted Nikon Eclipse Ti microscope equipped with a CFI Apo LWD Lambda S-series 20X Objective lens (N.A. = 0.9 ∞/0.11-0.23 WD = 0.95 mm WI Objective, Nikon) or 63x oil objective lens (N.A. = 1.4 ∞/ 0.11-0.23 WD = 130 µm, CFI PLAN Apo Oil Objective, Nikon). The samples were excited with 488 and 561 nm diode laser (LUN-F 100-240 V ~ 50/60 Hz, Nikon Instruments inc.), where channel acquisition intensities/exposure times used in the manuscript were as follows: 40%/100 ms (488 nm, 30 mW or 0.03 VA) or 25%/100 ms (561 nm, 11.5 mW or 0.015 VA) laser power settings for illuminating either endothelium or innate immune cells (macrophages or neutrophils) and Dextran, respectively. For 63x imaging, 488 nm laser excitation was reduced to ~10%/100 ms to maintain same laser light density during excitation (up to 6.8 W cm$^{-2}$). The fluorescence emission spectra were collected in 4D (XYZT) by automotive PCI hardware triggering (PXI-1033, National Instruments) with a Yokogawa CSU-W1 Spinning Disk unit (Nipkow disk pinhole = 50µm, 4000 rpm), incorporating a Photometrics Prime BSI Scientific CMOS (sCMOS) camera with analog Z-plane acquisition (MCL Nano-Drive NIDAQ Piezo Z, Nikon) and a motorized XY-stage (TI-S-ER, Nikon). The emission was collected for either green transgenic (Ex/Em:488/525; Gain: Correlated Multi-Sampling) or red Dextran (Ex/Em:561/605; Gain: High Dynamic Range) with a high-sensitive sCMOS camera using band-pass dichroic mirrors (Chroma Technology Corp., 89100bs) for filtering two separate fluorescence emission spectra (525/36 and 605/52, Chroma Technology Corp., 89000 Sedat Quad) placed in front of the detector for isolated detection of green and red fluorescence, respectively. The imaging plane (1331 × 1331 µm x 100 µm) was collected at 3 µm z step size to result with 35 slices and repeated with 30-second intervals per position for up to 60 min. Collectively, the live larvae were exposed to 5–6.81 W cm$^{-2}$ and 2–3.1 W cm$^{-2}$ laser power density for 488 and 561 nm, respectively. For all imaging experiments, the imaging dish was covered with a solution of interest at 270–300 s (t = 10-11$^{th}$ frame) with a bolus of 10x agarose volume ( ~ 2 mL). The z-stack images of zebrafish larvae were captured using multiple scanning modes at 100 µm tissue depth in z-axis at a resolution of 2048 × 2048 pixels (16 bit) in the x and y plane, corresponding to 0.645 µm/pixel calibration with a voxel size of (0.65 × 0.65 × 3 µm) in x y and z, respectively. The high-resolution XYZ-T image files were pre-processed by (2 × 2) binning and acquired as (1024 × 1024 per z-plane) with corresponding 1.538 pixel µm$^{-1}$ acquired image size (665.6 × 665.6 ×100 µm) through triggered excitation and emission collection for either red or green fluorescence in the NIS imaging software (NIS Elements, 5.12.0). Under these microscope settings, endothelial responses were imaged with a high spatiotemporal resolution where each 3D stack (frame) took

approximately 14 s (4 Z-stacks per minute) allowing multi-position acquisition of two transgenic or four dextran-perfused larvae in parallel in a single imaging experiment for 20x objective magnification.

## Spinning disk confocal microscopy and laser Wounding

For laser wounding experiments, single intact and anesthetized 3 dpf Tg (mpeg1.1:Qf2;Quas:cPla$_2$-mKate2-P2A-eGFP-KDEL) or 2-3 dpf Tg(kdrl:Qf2;Quas-cPla$_2$-mKate2-P2A-eGFP-KDEL) were embedded in 60 mm plastic petri dish (Corning, 351007) and immobilized with ~200 µL 1% LM agarose ISO$_{NaCl}$ ($\Pi$=280 mOsm) E3. Subsequently, the LM-agarose was enveloped with ~2-3 mL of hypotonic ($\Pi$=10 mOsm) or ISO$_{NaCl}$($\Pi$=280 mOsm) E3 media for creating a submerging environment for the 25X Objective lens (N.A. = 1.1 ∞/0-0.17 WD=2µm Water Dipping Objective, Nikon) in Nikon Eclipse FN1 upright microscope. The samples were excited with 488 and 561 nm diode laser lines (Andor Revolution XD) and fluorescence channel excitation intensity/exposure was adjusted to 35%/80 ms(488 nm) and 30%/80 ms(561 nm) for illuminating macrophage endoplasmic reticulum (KDEL-eGFP) and cPla$_2$-mKate2 in the caudal region of the larvae. The emission spectrum were excited in 4D (XYZT) by SmartShutter controller (Sutter Instrument, LB10-B/IQ Lambda) triggered excitation carried by Andor Laser combiner (LC-501A, Andor Technology) housing Andor iXon3 897 thermoelectrically cooled EMCCD camera with analog Z-plane acquisition (STG-STEPPER-Piezo Focus, Ludl Electronic Products). The emission was collected with a Yokogawa CSU-X1 spinning disk unit (Nipkow disk pinhole = 50 µm, 10,000 rpm) for green (Ex/Em:488/525) or red (Ex/Em: 561/625) in electron-multiplying mode (Gain: 10 MHz at 14-bit; Multiplier 300, conversion 1.0X) and band-pass filters for green (525/40, Semrock., FF02-525/40-25) or red (617/73, Semrock., FF02-617/73-25) fluorescence emission spectra. The imaging plane (287 X 287 × 50–80 µm) was collected at 1.5 µm z step size and repeated in no-delay intervals per position for up to 30 min, resulting with temporal step resolution of ~7–12 s per stack. The z-stacks were acquired at a resolution of 512 × 512 pixels (14-bit), corresponding to 0.561 µm/pixel calibration with a voxel size (0.56 × 0.56 × 1.5 µm) in x y and z, in the NIS imaging software (NIS Elements, 3.22.14). Collectively, live unwounded larvae were exposed to 5 W cm$^{-2}$ and 4 W cm$^{-2}$ laser power density for 488 and 561 nm, respectively. The wounds were induced at ~1 min (t = 5–7), with several successive laser pulses targeted at peripheral sentinel macrophages or nearby melanocytes for wounding in a single frame with 60 ms delay using microscope-mounted 435 nm ultraviolet Micropoint Laser (Andor), resulting in tailfin fin fold ablation around the boundary of hematopoietic region in the trunk of the larvae.

## Multiphoton imaging and FRAP-wounding

3dpf intact larvae Tg(mpeg1.1:Qf2;Quas: cPla$_2$-mKate2-P2A-eGFP-KDEL) were mounted in 35 mm glass-bottom microscopy imaging dish (MatTek Corporation, P12G-1.5-14 F) and transferred to Ti2-E inverted Nikon microscope, housing AX-R multiphoton modality (AX-MP NDD) and stage top incubator set to 28 °C (TOKAI-HIT, STX). The multiphoton images were excited through simultaneous absorption of 920 nm tuneable and 1045 nm fixed two photon lasers (mks, spectraphysics) for eGFP and mKate2, respectively. The excitation was adjusted to 920 nm (4.3%, Gain=45, Line Averaging=4X) and 1045 nm (5.3%,Gain=45, Line Averaging=4X) for eGFP and mKate2, respectively. The fluorescence emission was collected using 40X (Plan Apo lambda S 40XC, NA = 1.25 ∞/0.13-0.21WD = 300µm, Nikon) silicone objective with objective heating mantle (28 °C) and detected with PMT GaAsP. The bidirectional images were acquired in resonant scanning mode. The imaging plane (147 × 90 × 15 µm) is collected at Nyquist optical resolution of 0.149 µm at 1.5 µm z-step size using NIDAQ piezo and repeated in no-delay intervals for up to 2 minutes, resulting with a temporal step resolution of 2.5 seconds per stack. The z-stacks were acquired at a pixel resolution of 1024×626 (14-bit), corresponding to 0.144 µm/pixel calibration with a voxel size (0.14 ×0.14 ×1.5 µm) in x y

and z in the NIS imaging software (NIS elements, 6.02.01). The wounds were induced at -1 minute with a single high intensity 920 nm laser pulse (22%, Line Averaging=1X, Dwell time=1 s, area= $20 \times 20\,\mu m$) with photostimulation MP STIM FRAP modality in Galvano scanner mode, resulting in abrupt epithelial tailfin fold laser ablation around the boundary of hematopoietic region in the larvae. Furthermore, larvae were prepared similarly and also wounded in Leica stellaris 8 DIVE multiphoton microscope, equipped with two tunable IR excitation lasers: Mai Tai HP (690-1040 nm, continuous) and insight X3 (680-1300 nm, continuous). Within the Leica microsystem, tissues were wounded using the FRAP ab1 XYZT module in resnonant scanning mode with bidirectional scan direction, speed= 8,000 Hz and pixel dwell time 0.0319 μs. The images were acquired with HC PL IRAPO 40x/1.10 WATER, with 2.25x zoom at Nyquist XY pixel size 0.11 μm pixel$^{-1}$ resolution with a voxel size ($0.129 \times 0.129 \times 1.5\,\mu m$) in x y and z, corresponding to $1184 \times 1184$ (14-bit) and stack depth of 15 microns. For dual color excitation, the MAI TAI HP was tuned to 924 nm (MP1, 4.5%, power) for eGFP excitation and the insight X3 laser tuned to 1118 nm (MP2, 2.0% power) for mKate2. The wounding was performed using the FRAP module with a sequence of 4 prebleach iterations, followed by 6 bleach and 20 post-bleach iterations. For laser ablation, the bleaching intensity within ROI (area= $20 \times 20\,\mu m$) was set to 100% on MP2 (1118 nm) and 100% on MP1 (924 nm) in resonant scanning mode. We were not able to switch galvanic and resonant quickly within the same experiment with Leica stellaris 8 DIVE instead to induce efficient wounds at 8000 Hz scanning speed, WILL laser was also added at 100% and tuned to UV range with a total dwell time set to 2 s at mid-plane of the XYZT stack. Detection was performed using external spectral non-descanned hybrid detectors (NDD1 and NDD2) operating in photon-counting mode, with NDD1 configured for eGFP detection (480–527 nm) and NDD2 for mKate2 (620–732 nm). All imaging data were saved and processed using Leica LAS X software (Leica LAS X, 4.8.1.29271).

## Denoise and 3D deconvolution

The 4D image stacks of tg(mpeg1.1:Qf2;Quas: cPla2-mKate2-P2A-eGFP-KDEL) were corrected by 3D deconvolution, where sample PSF was derived and computed automatically with depth-calibration and without image intensity subtraction or preprocessing in Nikon NIS elements (5.21.03, Build 1489). The deconvolution process was carried by default method using landweber algorithm for spinning-disc modality with 50 μm pinhole size with immersion refractive index 1.33 (water) for both 488 and 561 nm fluorescent light emission[82]. The deconvolved or MP image stacks are denoised using denoise.ai tool for improving signal-to-noise in Nikon NIS elements (5.21.03).

## Generation of zebrafish CRISPR mutants

To generate zebrafish mutants a CRISPR/Cas9 system with a single sgRNA (Alox12-sgRNA1) targeting zebrafish alox12 (ENSDARG0000 0069463) exon7 is used. The ribonucleoprotein complex consisting of alox12-sgRNA1 and Cas9 recombinant protein is injected into the cytoplasm of one-cell stage fertilized zebrafish embryos. The injected F0 larvae were brought to adulthood and crossed with wild-type adults to produce F1 progeny. The alox12-sgRNA injected F1 zebrafish were grown to sexual maturity and their genomic DNA was isolated from their tail fins for genotyping. Tail fins were partially amputated, suspended in 250 μL of 0.05 M NaOH, incubated at 95 °C for 10 min, cooled on ice for 10 min, then neutralized with 25 μL of 1 M Tris-HCl (pH 8), and vortexed. The DNA extracts were used as genomic templates for Polymerase Chain Reaction (PCR). 486 bp PCR products were digested with FastDigest BseLI (Thermo Fisher Scientific, FD1204) overnight at 37 °C, producing two DNA fragments (310 and 176 bp). The 508 bp represents a mutant allele where the BseLI site has been disrupted by Cas9-induced mutation. The 508 bp band was isolated from the agarose gel and sequenced via Sanger, confirming a 22 bp

insertion (Supplementary Fig. 2g-h, 2k). Subsequently, the F1 heterozygous adult zebrafish with 22 bp frameshift mutation were bred to homozygosity.

Two independent sgRNAs (lta4h-sgRNA-1 and lta4h-sgRNA-3) targeting zebrafish lta4h (ENSDARG00000006029) were used to establish a CRISPR-zebrafish line[26]. The Cas9-gRNA ribonucleoprotein complex (a combination of lta4h-sgRNA1 and lta4h-sgRNA3) was injected into the cytoplasm of one-cell-stage zebrafish embryos. After the injected F0 larvae matured (2-3 months post-fertilization), individual F0 adults were crossed with wild-type adults to produce F1 progeny. These F1 larvae were then grown to sexual maturity, and genomic DNA was isolated from their tail fins for genotyping. For lta4h-sgRNA1, PCR products were digested with FastDigest BseLI (Thermo fisher Scientific; FD1204), producing three DNA fragments (271, 140, 129 bp). For the lta4h-sgRNA3, PCR products were digested with FastDigest SalI (Thermo fisher Scientific; FD0644), producing three DNA fragments (271, 184, 85 bp). The 271 bp product represents a mutant allele where the BseLI or SalI site has been disrupted by Cas9-induced mutation. This 271 bp band was isolated from the agarose gel and sequenced via Sanger sequencing, confirming a 5 bp deletion (2, 2, 1 bp deletion) and a 7 bp insertion (Supplementary Fig. 2i-j, 2m). F1 heterozygous adult zebrafish with this frameshift mutation of interest were bred to achieve homozygosity.

A single guide was designed targeting zebrafish pla2g4aa (ENSDARG00000024546) for genetic disruption at exon 7. The Cas9-gRNA ribonucleoprotein complex consisting of pla2g4aa-sgRNA1 was injected into the cytoplasm of one-cell-stage fertilized embryos. The injected larvae were then brought to adulthood and the individual F0 animals were used to produce F1 progeny by backcrossing with wild-type Casper fish. The F1 larvae were grown to sexual maturity, genomic DNA was subsequently isolated from their tail fins and sequenced. The pla2g4aa$^{wt/mk220}$ DNA was PCR amplified, digested with FastDigest BseMI (Thermo fisher Scientific; FD1264) and then analysed by agarose gel electrophoresis for genotyping. The 447 bp PCR product from the wt pla2g4aa allele is cleaved into two smaller products upon overnight incubation with BseMI (281 and 166 bp) while the 437 bp PCR product of the mk220 allele is not cleaved by restriction digest into the smaller products. F1 heterozygous adult fish with the mk220 10 bp deletion (Supplementary Fig. 5a-b, 5e) were bred to achieve homozygosity. The Genomic exon-intron architectures (Supplementary Fig. 2g, 2i, 5a) were obtained from the Ensembl database for Danio rerio (GRz11 Genome assembly; GCA_000002035.4). The exon–intron schematics were downloaded and adapted (recoloured) for display visualization of exon targeting[83].

Alox12 (Chr 7, alox12 Ensembl: ENSDARG00000069463) mk218 allele Exon7-ins-22bp

 gRNA2: GGATCACTGGGCAGAAATAC TGG

 Fwd: 5'-CAAATGCATTGATGCAAAAAGT-'3

 Rev: 5'-TGAGAAATAGCATTCATTTGCG-'3

Lta4h (Chr 4, lta4h Ensembl: ENSDARG00000006029) mk219 allele Exon1-ins-7bp-Δ5bp

 gRNA1: GAAAGTCGCCCTGACTGTGG AGG

 gRNA3 CATGCCTGTCAAAGTCGACA TGG

 Fwd: 5'-TCAACCATGACTCCAGTTTCAG-'3

 Rev: 5'- CAGTGCATTGGATCGTACTCAT-'3

cPla2 (Chr 2, pla2g4aa Ensembl: ENSDARG00000024546) mk220 allele Exon7Δ10bp

 gRNA1: GGAGGTTTTCGTGCAATGGT GGG

 Fwd: 5'-CCATCCTCACCAAGAGAGGTAA-'3

 Rev: 5'-ACTGCTTGAATTGACTGCAAAA-3'

## Larval genotyping of established zebrafish CRISPR mutants

2.5, 3 or 4dpf Zebrafish Larvae were recovered from the 35 mm imaging dish and subjected to 20 μL 0.05 M NaOH alkaline DNA extraction at 95 °C and neutralization with 2 μL 1 M Tris-HCl (pH 7.4)[26]. The gRNA

targeted exon was PCR amplified and subjected to restriction enzyme digestion followed by gel electrophoresis analysis. For imaging experiments where the genotype was unknown (in-cross of hetero-zygotes for pla2g4aa), blinding was inherent in the experimental design and the genotypes were matched to the assigned fish *post hoc*. For all genetic experiments supporting the findings of this paper use zebrafish larvae at age 2.5-4 dpf. The previously published and estab-lished zebrafish lines were genotyped using the following primers:

Alox5a (Chr13, *alox5a* Ensembl: ENSDARG00000057273) mk211allele Exon7Δ10bp[36]

gRNA1: TGGGTGCCGCCAAGTACTGA TGG
Fwd: 5′-GCTGTAATCCAGTGGTCATCAA-′3
Rev: 5′-TGATCTCACTGGAGACTGGAGA-′3
Hcar1-4 (Chr12, Ensembl: ENSDARG00000087084) mk214 allele Exon2 4ins/ Δ 756 bp[26]

gRNA2: GGTAAAGGATCCTGAAGAAG CGG
gRNA4: GGCATGGAGACACACAATGA GGG
Fwd: 5′-TGCCTAAACATTTGTGTTCGTGT-3′
Rev: 5′- AGACTGCCGAATGTTGGTGT-3′

## Chemogenetic depletion and Sudan Black staining

To perform macrophage-specific cell depletion, Tg(*mpeg1.1*:Qf2;5x-Quas:GAP-tagYFP-P2A-NfsB_Vv) animals tagged with nitroreductase under the mpeg promoter were used. Tg(*lyz*:Qf2;5xQuas:GAP-tagYFP-P2A-NfsB_Vv) was used as a genetic vehicle control. The 100 mM MTZ (Sigma-Aldrich, 1442009) stock solution was prepared in DMSO (Mil-lipore Sigma, 276855) and protected from light. Embryos were dechorionated prior to metronidazole (MTZ) ablation using 1 mg/mL pronase (Roche, 165921). Starting at 2 dpf, the larvae were treated with 150 μM MTZ and 0.15% DMSO in hypotonic ($\Pi$ = 10 mOsm) E3 media or mock-treated with 0.15% DMSO in hypotonic E3 as a vehicle control for at least 24 hours. At 3 dpf, MTZ was temporarily removed and replaced with hypotonic ($\Pi$ = 10 mOsm) E3 media until the microangiography procedure and subsequent immobilization in $ISO_{NaCl}$($\Pi$ = 280 mOsm) low-melting agarose for confocal imaging. During confocal imaging, larvae were shifted back to hypotonic ($\Pi$ = 10 mOsm) E3 media con-taining either 150 μM MTZ or 0.15% DMSO, depending on their respective treatment conditions. For pharmacological inhibition of eicosanoid pathway, macrophage depleted or DMSO treated larvae were both inhibited with 30-45 min 50 μM Licofelone dissolved in hypotonic ($\Pi$=10 mOsm) E3 and subjected to dextran micro-angiography and imaging. For Sudan Black staining, (Tg(*mpeg1.1*:Qf2;Quas:NTR2.0)) or (Tg(*lyz*:Qf2;Quas:NTR2.0)) larvae were used for MTZ or DMSO vehicle treatment regiment for 24hrs, followed by wounding and fixed at 90 min post wounding using 4% paraformaldehyde (Fisher Scientific, BP531) in 1x PBS overnight at 4 °C and then stained with Sudan-Black for 30 min. The larvae were rinsed three times in 70% ethanol (Decon Labs, 2405) and rehydrated with PBS-Tween-20 for 5 min. Prior to imaging, the larvae were depig-mented (1% KOH, 1% $H_2O_2$) and washed for total of three times in PBST solution 5 min each and transferred for transmitted light imaging[26].

## Image processing and measurement of interstitial serum leak-age and Dilation

Nikon ND2 Spinning disk confocal time-lapse stacks of wounded larvae were imported into Fiji (v1.54 J Just ImageJ Package) using bio-formats plugin and z-projected (Maximum-Intensity Projection) with brightness and auto-contrast. To minimize the movement of the imaged plane, the 3D (XYTZ = MAX) was registered using 2D Linear Stack Alignment SIFT multichannel (PTBIOP plugin) tool and the dextran channel was set as reference on the imported composite image stacks (Scale Invariant Interest Point Detector) and rigid transformed (maximal alignment error:50px, inlier ratio:0.05) for subsequent morphology or intensity analysis of endothelium or and dextran intensity and endothelial tailfin ratio. For genetic experiments, the registration was carried out using

SIFT Linear Stack Alignment with default settings. For quantification of serum leakage, dextran intensity was traced per larva in a stratified manner in the extra-epidermal 20-50 μm wounded region ($I_{bleed}$(t)) and below the caudal arteries & veins ($I_{vessel}$(t)). Both Iv and Ib datasets were subtracted with background (empty area in field of view) intensity per timepoint for correcting the noise of sCMOS camera, except for 500 kDa imaging as the complete FOV does not capture extraepidermal dextran from the tailfin tissue wound in 2.5-4 dpf zebrafish larvae. For pharma-cological experiments, quantifications were carried out with PTBIOP plugin. For ratio measurements of endothelium and tailfin, a custom script was developed in python, using scikit and numpy packages that allows filteration of small objects or cells in the epidermis and detection of single continuous binarized ( = Otsu) endothelial signal and tailfin area based on 70 kDa dextran and calculated as ratio in FOV, the image was loaded and analyzed in napari. The quantifications were carried out manually to monitor the accuracy of defined registered regions in the tailfin using set rectangle tool with fixed dimensions for wound (50 × 50 pixels), vessel (300 × 100 pixels), respectively and quantified using plot z-axis profile function for all available time-frames for both 70 kDa and 500 kDa Dextran(ImageJ, fiji). For endothelial morphological parameter measurements and dilation, vertical line was drawn using line tool with fixed dimensions overlaying the split and registered endothelial fluor-escence signal channel (length=50 pixels, with=30) and multi-kymograph function with default settings. The resulting kymograph was binarized (Threshold method=Otsu, Dark Background) to create an 8-bit image and area quantified using makeline tool (width=30 pixels, length=121) and the diameter measured using the plot profile function.

To ensure comparability of results obtained across different experimental days and injection mixtures number of normalization methods were used. For systemic statistical comparisons, vessel dila-tion, vessel leakage and wound permeability were quantified using normalized formulations.Vessel dilation ($D_v$(t)= vessel diameter, Eq. 1), wound permeability ($I_b$(t)= wound leakage, Eq. 2), and vessel perme-ability ($I_v$(t)= vessel leakage, Eq. 3) were quantified by normalization to baseline (0–270 s) and expressed using the following mathematical formulations.

$$D\mathrm{vnorm}(t) = \frac{Dv(t)}{\sum_{t=0s}^{t=270s} Dv(t)} \tag{1}$$

Vessel dilation (Eq. 1): Expressed as the relative change in vessel diameter over time, normalized to the cumulative baseline diameter measured during the first 270 s. This accounts for inter-experimental variation in baseline vessel size.

$$I\mathrm{bnorm}(t) = \frac{Ib(t)}{\sum_{t=0s}^{t=270s} Ib(t)} \tag{2}$$

Wound leakage (Eq. 2): Reported as the relative wound intensity over time, normalized to the cumulative baseline wound intensity (0–270 s). This allows comparison of leakage dynamics independent of initial wound intensity values.

$$I\mathrm{v(t)tot/norm} = \frac{\int_{t=0s}^{t=3600s} Iv(t)}{\sum_{t=0s}^{t=270s} Iv(t)} \tag{3}$$

Total vessel leakage (Eq. 3): Calculated as the time-integrated vessel intensity over the full imaging window (0–3600 s), normalized to the cumulative baseline vessel intensity. This metric captures the overall leakage while correcting for baseline fluorescence variation.

The intensity measurements are reported as arbitrary values (arb. units). All imaging data was normalized to the first 10 timepoints per imaged larva using Python (Conda, v3.8.10 64-bit) and MATLAB (R2024a, v24.1.0.2537033, 64-bit) code and combined, except for leakage rate measurements. Leakage rates were derived from temporal

derivatives of the intensity signals. The apparent wound leakage rate was defined as ($Rb' = \frac{dIb}{dt}$) representing the rate of change in wound intensity over time. The vessel leakage rate ($Rv'$) was calculated from the corresponding vessel intensity dynamics.

Mathematically, if vessel intensity remains relatively stable or constant over the imaging period, then the area under the curve integration simply reflects the baseline signal scaled by the experiment duration (3600 s). Thus, Eq. (3) produces a normalized value of 3600, which can be interpreted as the expected hypothetical outcome in the absence of leakage. Deviations above this level indicate cumulative leakage relative to hypothetical baseline. Note that statistical analyses were performed without subtracting the fixed baseline integration value of 3600. The image acquisition and image analysis dimensions are set to be consistent between all samples, and these measurements are designated above in the microscopy and analysis section and independently represented. The 2D and 3D image analysis was performed with Anaconda distribution of Python (Python ≥ 3.8.10). Specifically, customized python scripts were developed using the Numpy (v.1.23.5)[84], SciPy(v1.15.2)[85] scikit-image (V0.19.1)[86], matplotlib (v3.5)[87], trackpy (v0.6.4)[88,89], Allen Cell Structure Segmenter (v0.5.0)[90] and napari (≥ v0.4.10)[91] libraries. GPU acceleration (when available) was provided by CuPy (v12.0)[92].

### 500 kDa dextran onset of permeability quantification and kymograph area

Nikon ND2 Spinning disk confocal time-lapse stacks of wounded and non-wounded larvae were imported into Fiji (v1.54 J Just ImageJ Package) using bio-formats plugin and z-projected (Maximum-Intensity Projection) with brightness and contrast-enhanced. the registration was carried out using SIFT Linear Stack Alignment with default settings. Similar to 70 kDa processing (Eq. 3). Furthermore, to trace the high-molecular weight dextran, the imaged stacks were loaded and analyzed for onset of leakage). The dextran channel was min-max normalized and projected for 1024×1024 pixels and collapsed 121 frames to form kymograph. The endothelial contour was segmented and overlayed with turbo look up table to correlate thresholded pixel position through time. The obtained mask is subsequently loaded for area analysis $Iv_{Area}$ (arb. units), where complete FOV area is $1.048578 \times 10^6$ pixel$^2$ or $4.43 \times 10^5$ μm$^2$. Unfortunately, we are not able to detect extraepidermal wound leakage with 500 kDa as it is absent outside tissue at baseline wound condition (Eq. 2). $Iv$ statistical and comparative analysis was conducted similarly to 70 kDa dextran between matched controlled conditions.

### Image-based measurement of cPla$_2$ nuclear membrane binding in macrophages and endothelial cells

The 4D spinning disk confocal stacks were imported into napari (v0.4.10). Raw images were intensity-normalized and smoothed using a Gaussian filter, background fluorescence along the z-axis was subsequently subtracted using a rolling-ball algorithm. The cPla$_2$-mKate2 channel was segmented in 3D using the watershed algorithm and middle frame is isolated based on the largest contour area and seeded for segmentation analysis. A standard particle tracking algorithm was employed to track the nuclei movement over time and the non-track objects were cleared. cPla$_2$-mKate2 membrane binding was defined as the ratio between mKate2 intensity at nuclear rim and nucleoplasm at the middle frame. The obtained 3D tracks were directed to normalization and edge-preserving smooth function. The obtained tracks were quantified for nucleus volume geometry and protein binding over time and the discontinuous tracks were excluded from the analysis. All frames among different remaining tracks were unionized and cPla$_2$-mKate2 binding ratio at subsequent timepoints are normalized to cPla$_2$-mKate2 intensity in the 1$^{st}$ frame of the image stack.

### cPla$_2$-mKate2 emission and 2-D quantification

The Tg(mpeg1.1-cPla$_2$-mKate2) emission was quantified for signal distribution in 2D using Fiji (v1.54 J Just ImageJ Package). The quantifications were conducted on isolated macrophage Images that were MIP processed and duplicated from whole field view stacks (XY-T) and registered using SIFT Linear Stack Alignment with default settings. The obtained images were quantified using "line" tool and "Plot Profile" function by drawing line-plots on the periphery of macrophage nuclei and quantifying its distribution over set time-points. The obtained data were then plotted for analysis of transient cPla$_2$-mKate2 localization.

### Quantification of leukocyte recruitment in Sudan Black Staining

The still images of the Sudan-black stained larvae were processed in Fiji with z-project function (Maximum-Intensity Projection), duplicated and blurred (Gaussian, sigma= 50 scaled). The background subtracted using image calculator function ("Subtract create 32-bit", ""), resulting 32-bit floating image was median-filtered (radius= 5) and quantified for number of leukocytes in the tailfin using find maxima tool (prominence= 1950 strict light) and counted. The counts were verified for accuracy by manual counting and analysed.

### Time-lapse neutrophil and macrophage tracking quantification

For analyzing the migration patterns of neutrophils and macrophages to osmotic wounds, we developed a custom python script that relies on python 3.9.23, housing skimage, scipy numpy, napari and matplotlib plugins for analysis and segmentation of moving cells in (Tg(mpeg:Qf2;Quas: NTR2.0)) and (Tg(lyz:Qf2;Quas:NTR2.0)). In brief, approximately 10 parameters were defined within a single script to isolate the tracks of neutrophils and macrophages. The binarized cell channel is filtered per-frame and tracked for nearest-neighbor algorithm and to correct for ID assignment, centroid-tracking algorithm was implemented for fallback ID re-assignment of original tracks ("Ghost Tracks"). The obtained data was unionized for filtering redundant tracks, and filtered data is used to generate net object displacement for per-object and averaged across biologically independent experiments. For macrophage, the centroid-based tracking was not consistent due to amboeid movement. To isolate macrophage tracks, a hybrid of fallback and nearest neighbor algorithm that relies on hybrid of polygon and centroid-based tracking was used. Each object angular direction information was also extracted for rose polarity plots. The obtained angular data was extracted as angle_rad from biological independent experiments, averaged and plotted as rose plots for neutrophils and macrophages, respectively. Some of the files produced large amount of fragments due to centroid or polygon based tracking, those files that had excessive cell-cell interaction during the course of the imaging experiment, were not analyzed for tracks parameters.

### Statistics and reproducibility

No statistical methods were used to predetermine sample size and the sample size is similar to those reported in previous publications[4,36] in brief we always used sample sizes of 8 different animals or more in the study, for laser wounding 3 different animals or more. Unless otherwise indicated, the data analysis was performed blind to the condition of the experiments and each experiment was repeated at least twice with similar results. The specimens in the imaging experiments were randomly allocated to experimental groups without any bias or pre-selection. Investigators were not blinded to allocation except for genetic experiments with pla2g4aa where all the measurements were taken prior to the authors knowing the genotype status of the larvae. The data was sorted into groups post genotyping. Blinding was not performed for the sake of increasing experimental throughput. The calculations in Fiji and napari were completed before calculations in MATLAB and significance quantification in Prism. The statistical analysis was completed using GraphPad Prism 10 (version 10.2.3). Each

dataset was tested for normal distribution (Shapiro-wilk or D'agostino-Pearson) test. Only if the data were normally distributed, a parametric method (unpaired two-tailed students t-test) was applied to the dataset. A non-parametric test (two-sided Mann-Whitney) was applied for non-normally distributed data sets. In case of multiple comparisons, one-way ANOVA or Welch one-way with Dunnett's multiple comparison post-hoc test applied. Kruskal–Wallis testing, a nonparametric alternative to one-way ANOVA, was used due to non-normal data. For analysis with different variance in error distribution, Welch's correction was used. Data from two groups were compared using two-tailed unpaired $t$-test. $P < 0.05$ were considered significant. Data were represented as bar plots where one dot in a graph represents $n = 1$ biologically independent larvae experiment (single larva). Dot plot annotated numbers represent the total $n$ per group at the top and mean value in the bottom of the graph. In bar plots top and bottom edges indicate SD range and midline central marking is representative of mean values. The Data distributions errors are shaded and represented as 95% confidence intervals, unless otherwise indicated. $*P < 0.05$, $**P < 0.01$, $***P < 0.001$, $****P < 0.0001$ and ns, not significant. For every pharmacological treatment, and control larvae were derived from the same spawn of embryos. For transgenic imaging experiments, larvae were derived from multiple independent clutch of embryos from the same day or more commonly independent clutch of embryos on multiple days. We did not include selective breeding approach for transgenic imaging epxeriments as these larvae derive from multiple parents per indicated date. Animals were never reused for different experiments and all statistical analyses were performed across biological replicates. Replication included variation in biological source conducted across several experimental days from independent clutch of embryos. Analysis was performed objectively without blinding.

The larvae were selected on the following criteria: normal morphology, a beating heart and circulating red blood cells. Data exclusion criteria were predefined. Animals were excluded if yolk and/or cardiac function did not restore after dextran injection experiments. The specimens were randomly chosen for pharmacological, chemogenetic genetic or transgenic imaging experiments without bias or pre-selection. Imaging datasets were excluded if affected by artificial fluorescence fluctuations (e.g., dust debris), incompletely acquired Z-stacks or experiments where tailfin tissue drift exceeded 300 pixels ( ˜150 microns) in the XY plane or 50 pixels ( ˜25 μm) in the Z-direction and were completely discarded, since no reliable data can be extracted. For x63 Oil objective imaging, the live zebrafish at 2.5-4dpf do not have flat tail morphology and the depth of objective does not always reach the sample. Furthermore, Nikon's Perfect Focus Function (PFS) was not able to focus an IR beam with the x63 objective, therefore most image data acquired drift upon shift in osmolyte or bathing medium, we have only retained data that do not show excessive XYZ drift in the imaging experiment. For each UV or IR-induced tissue wounding, directly damaged cells that were within the wound ROI ablation region were excluded from the analysis. No data was excluded based on distribution (no outlier exclusion). Details of statistical analyses used, exact P values, number of experiments and animals for all graphs are listed in figure or statistical source data file.

### Reporting summary
Further information on research design is available in the Nature Portfolio Reporting Summary linked to this article.

## Data availability
Data supporting this work are available in this Article. The data generated in this study are provided in the Supplementary Information or source data file. The demo image data generated in this study, custom software script, MATLAB script and Fiji-java script are available in the GitHub database under accession code (https://github.com/zazadovv/Macrophage_Vessel.git). All other data supporting the findings of this study are available from corresponding author upon request. Specifically, the time-lapse and multi-photon imaging data can be made available only upon request due to its proprietary file formats, data privacy laws, very large file sizes (Total>20TB) and permanent repository storage limitation. The primers are provided in the manuscript methods section, the plasmids and zebrafish strains created in this study are listed in methods section and are available in compliance with animal transfer agreements and institutional animal care IACUC approval. Source data are provided with this paper.

## Code availability
Analysis scripts and image and data processing pipeline are available on GitHub page for Fiji, Python and MATLAB. All relevant analysis scripts, demo files, environmental files and package information detail can be accessed on github (https://github.com/joeshen123/Nuclear-Membrane-Binding-Analysis.git; https://github.com/zazadovv/Macrophage_Vessel.git)[93].

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

## Acknowledgements

We thank Deji Afolalu for aquatic facility management, Mariia Akhmanova and Tasos Siskoglou for their expertise in multiphoton imaging and Denis A. Larochelle, Mariia Akhmanova, Miklós Lengyel and Srivatsan Rajan for valuable comments on manuscript. This research was supported by the NIH/NIGMS grants R35GM140883 to P.N. and in part by the NIH/NCI Cancer Center Support grant P30CA008748 to P.N. and Gerstner Sloan Kettering Harold E. Varmus Fellowship (Mr. and Mrs. Rosen Family) to Z.G.

## Author contributions
P.N. and Z.G. conceived the study and designed the experiments. Z.G. and Z.S. conducted the experiments. Y.M., M.J. and Z.G. designed and established the new CRISPR genetic and transgenic lines in the study. Z.S. and Z.G. developed the Python and Fiji image analysis scripts and analysed the data. Z.G. prepared the figures. Z.G. and P.N. designed MATLAB scripts for data analysis aided by ChatGPT4.0. Z.G. and P.N. wrote the paper.

## Competing interests
The authors declare no competing interests.
