## [Transparent Peer Review file · Nature Communications]

Zebrafish Macrophages Convert Physical Wound Signals into Rapid Vascular Permeabilization

Corresponding Author: Dr Philipp Niethammer

Version 0:

Reviewer comments:

Reviewer #1

(Remarks to the Author)

In this article, Gelashvili and colleagues investigate the mechanisms by which wounds are detected and responses are induced, using Zebrafish as a model system. They use fluorescent dextran in the blood circulation to measure leakage from the blood vessels and from the wound. Careful quantification and use of the right controls shows that leakage from the blood vessels is induced by the osmotic shock accompanying the wound and requires Alox5a, suggesting a role for the cPLA2 pathway. The authors then show that immune cells, more specifically macrophages, are needed to induce leakage of dextran from blood vessels upon wounding, and that cPLA2 is translocated to the nuclear envelope in macrophages rapidly after a wound; they also show that cPLA2 mutant animals are defective in the induction of leakage by wounding. They propose a working model by which osmotic stress induced by the wound rapidly induces swelling of cells and nuclei in the tissue, leading to cPLA2 activation in macrophages, production of arachidonic acid by these cells, which promotes blood vessel leakage via Alox5a. Overall the article is very clear, contains carefully executed experiments and controls, and provides a very nice demonstration of how the activation of cPLA2 in immune cells contributes to the response of tissues to wounds. I would thus recommend publication of the article. I have just a few minor concerns/questions.

- 1) The authors write that they observe recruitment of cPLA2 to the INM of macrophages, referring to figure 4. While I understand that other articles demonstrate well this recruitment, it is not clear in figure 4. Maybe the authors mean enrichment at the nuclear periphery? This is quite clear from the quantifications, but it is hard to see how the INM can be distinguished here.
- 2) In most perturbations, the amplitude of the reduction in dextran leakage appears quite small. The authors claim that the pathway they identified is 'required' to induce this leakage. Maybe they could rather say that it 'contributes'. It is not clear for me (I am not a specialist of this phenomenon nor these techniques), whether the reduction they observe would correspond to what is expected in the case of a total lack of leakage induction, or it is really just a partial effect, meaning that the cPLA2 pathway in macrophages is just one of the signals inducing this leakage. This should be clarified in the discussion of the results (but it could also be that I misunderstood something here).
- 3) The authors showed several elements of their working model independently of each other. If I understood correctly, they showed that macrophages contribute to the response, that cPLA2 also contributes (although the depletion is in the whole animal and not specifically cPLA2) and that they observe cPLA2 enrichment at the nuclear periphery in macrophages (but they did not seem to have stained other cell types). So they never test directly a role of cPLA2 in macrophages, and they also do not exclude a contribution of cPLA2 in other cells (like endothelial cells) - is cPLA2 also recruited to the nuclear periphery in endothelial cells, or in other cells of the tissue? Can they deplete cPLA2 in specific cell types? If more specific experiments are not feasible, I still think that their working model is not contradicted by their experiments, but they might want to tone down their conclusions and make it clear that there are other possible contributions to the induction of leakage upon wounding (like cPLA2 activation directly in other cell types, or indirectly, and the fact that there is no direct proof of the contribution of cPLA2 specifically in macrophages). Again, I apologize in case of misunderstanding on my side - this system is quite complex and I might have missed important elements of proof for the working model proposed.

(Remarks on code availability)

Reviewer #2

(Remarks to the Author)

In this work, Gelashvili et al., presents evidence that macrophages localizing to the perivascular space of embryonic zebrafish blood vessels trigger a quick response to wounding by regulating osmotic signals and vascular permeability. The work addresses an interesting and important aspect of wound healing and introduces a novel signaling pathways into the mechanism of tissue injury. The article is written in clear and detailed language, the data is of good quality and quantifications were carefully performed, however, there are several conclusions that would require further validation to support the author's conclusions more directly, for example, a characterization describing the vascular phenotypes of the mutants described in this study under homeostasis would be a key point to interpret the results presented by the authors, as this would provide insights into the mutants vascular integrity, architecture and function. Some of my concerns and questions are described below:

Figure 1

- Is there also response to the wounds caused by intracardial injection? Even if this delivery is done under strict care, this procedure should still cause a wound on the site of puncture, therefore, a wound response may be seen at this site as well.
- Dextran tend to diffuse to tissues outside the vasculature – which is evident in all figures – the amount of diffusion may vary from fish to fish based on artifacts like the injury induced at the injection site, or the minimal differences in tail clipping sizes, the authors should show non-wounded animals as controls and include the values obtained in their quantifications. Alternative, they could use fluorescent qdots or nanobeads which stay in circulation for longer periods of time without diffusing to adjacent tissues and have been extensively used to label vascular structures in zebrafish. Any diffusion would be strictly at the wound site which could facilitate a more restricted quantification, but also help visualizing vascular phenotypes much more clearly.
- It would be helpful to relabel this figure, assigning each panel a letter for easier reading, although this version is properly described, having single letters for each panel will make the manuscript easier to follow especially in figures with multiple panels.

Figure 2

- What is the vascular phenotype in the *alox5a*, *alox12* and *ita4h* mutants? The few vessels showing in this figures hint of vessels having significant architectural differences (figure 2b,c and d), where they appear much thinner and less branched or developed, potentially explaining why some of these mutants (*alox12* and *ita4h*) showed reduced vessel leakage. Is the vasculature in these mutants leaky in general? Once again, showing an uninjured embryo's vasculature labeled with qdots could provide insights into the vascular phenotypes of these mutants. This is also applicable to all the drugs quantified in Figure 2, where no embryo data is shown to support their vascular leakage observations.

Figure 3

- Based on references to zebrafish transcriptomics the author mention that *alox5a* and *ita4h* are expressed by immune cells – potentially macrophages and neutrophils, however they do not show a staining validating this, which makes their ablations informative but not conclusive on the specific role of these genes in vessel permeabilization – as many other factors coming from these immune cells could lead to their results. Similarly, expression of these genes by non-immune cells was not tested.
- As the authors have already made *alox5a* and *ita4h* mutants, they could repeat these experiments on these animals or test their hypothesis transplanting mutant macrophages/neutrophils into wild type animals and testing whether these mutant cells affect vascular permeability/leakage.
- Here the authors mentions that the NTR/MTZ dependent ablation of neutrophils did not modulate vessel permeability, it would be good to see examples of these treated animals and the general distribution of their immune cells comparing inside and outside the vasculature.
- It is also not clear how far are macrophages and neutrophils migrating in response to the wound and the change in osmotic solutions?

Figure 4

- Once again, showing the vascular phenotype of the *cPla2* mutants would be helpful to get a sense of the vascular architecture and integrity.
- The author mention 'perivascular macrophages' during their nuclear shape sensing analysis in *cPla2* transgenics, and elaborate on this in the discussion, it would be helpful to characterize these cells and determine whether these are classic *mpeg1+* macrophages localized outside vessels or actual perivascular immune cells with a specific cellular profile (e.g. *mrc1/lyve1+*)?
- A general comment is that many of these experiments were performed on the casper mutants, do these mutants have any alterations in the way wound healing occurs, or immune cells react to wounding? the authors should either show this mutant background does not affect their results (compared to a WT) or reference studies pointing out to previous reports where this was addressed.

(Remarks on code availability)

Reviewer #3

(Remarks to the Author)

The manuscript Gelashvili, et al, focuses on understanding how macrophages signal to respond to wound signals and initiate rapid vascular repair. Though the data is interesting, new insights since previously published work seem limited for

the scope of Nature Communications. The imaging is carried out at fairly low resolution and while the responses to changes in osmotic stress are pertinent, the phenotypes are modest and definitive links to mechanics and the proposed genetic pathway are lacking. The paper is overly technical to read, and somewhat hard to follow the logic of. I appreciate the immense amount of work presented; however, in this format, the paper is not well targeted for general readership.

(Remarks on code availability)

Version 1:

Reviewer comments:

Reviewer #1

(Remarks to the Author)

In their revised article, the authors have provided new data and analysis, including more controls and clarification of level of wound induced leakage, also being more subtle in their claims to leave open some alternatives when they did not provide perfect direct proofs - for example for the fact that the downstream products of the cPLA2 pathway are mostly coming from perivascular macrophages. They tried some direct transplantation experiments that failed, and as I am not a zebrafish expert, I unfortunately have no specific idea to suggest to go around this experiment to test their hypothesis more directly. I think that, despite this failure for a direct proof, their article provides very interesting and convincing data to support their working model, and that the way they now present their results and discuss them, is well adapted to the level of confidence they can have in their results. I would thus recommend publication without further revisions.

(Remarks on code availability)

Reviewer #3

(Remarks to the Author)

The included revisions have decreased my initial concerns, and dramatically improved the manuscript

(Remarks on code availability)

Reviewer #4

(Remarks to the Author)

I have been appointed to replace Reviewer #2, who are no longer available assess the authors' response to their previous comments.

After reviewing the revisions that the authors performed, I believe that the authors have in general satisfactorily addressed the Reviewer #2 comments.

However, I would like to raise a few points:

1. Given that the images of blood vessels are low in magnification and resolution, it is challenging to determine whether there is a vascular phenotype in the *alox5a*, *alox12* and *ita4h* mutants; and between controls and after osmotic shock. Based on this resolution, there is no observed gross differences at the tissue scale, as the authors claim, although the ventral caudal vein of *alox5a* mutant appears more dilated than the wildtype embryo to me (Supplementary Fig. 2).
2. I would have liked to see the effects of osmotic shock (after injury) on endothelial cell-cell junctions (morphology) and upon the loss of macrophages and/or perturbation of the cPLA2-ALOX5-LTA4H pathway. This will provide some insights into how blood vessel permeability is increased.
3. The authors stressed the function of nuclear membrane mechanotransduction in transmitting wound signals to blood vessels and emphasized the importance of inner nuclear membrane (INM) tension in recruiting cPLA2 to the INM after osmotic shock. The evidence supporting this claim is rather weak in this manuscript. This is partly because it is not possible to see the shape of cell nuclei as well as the shape of the macrophages, thus making it difficult for non-macrophage experts to determine the subcellular localization of cPLa2-mKate2. I would have appreciated a more detailed examination of cPLa2 localization with, for example, a nuclear/nuclear membrane marker to clearly show that cPLa2 redistributed from the nucleoplasm to the nuclear membrane. Along this line, why does cPLa2-mKate2 distribution look different in Fig. 4e (more elongated, like the cell body) compared to Fig. 4c (more circular)? Does HYPO condition change the distribution of cPLa2 or the shape of the nucleus?

(Remarks on code availability)

Version 2:

Reviewer comments:

Reviewer #4

(Remarks to the Author)

The authors have sufficiently addressed my comments as well as clarified my questions.

(Remarks on code availability)

Response to Reviewers:

We would like to thank the reviewers for their interest and thorough assessment of our manuscript. Based on their critique we revised the manuscript providing a substantial number of new experiments and improved figure/data organization/presentation. We also largely revised the results and discussion section for better balance and focus and reduced technical terminology.

The revisions include:

- Revised vessel leakage quantifications (baseline leakage now indicated and discussed) throughout the manuscript for easier estimation of perturbation effect sizes.
- Inclusion of non-wounded animals into the analysis as requested.
- New 500-kDa dextran assays, also at higher magnification (63x), in wounded and non-wounded animals along with an assessment of baseline vascular morphology/permeability as requested.
- Figures streamlined and relabeled to improve logical flow; a single overview cartoon now integrates genetic, pharmacologic, and chemogenetic results. Reduced technical terminology as requested.
- Cell-type specificity of cPla2 response: We now also test for osmotic cPla2 activation in the endothelium as requested.
- We added additional OXER1/Hcar1-4 mutant experiments to further delineate the molecular mechanism of osmotic vessel permeabilization (i.e., distinguish it from osmotic surveillance by leukocytes).

Altogether we believe that our extensive revisions significantly improved the overall strength, focus and accessibility of our findings.

Please find below a point-by-point discussion of all the comments.

Kind regards,

P. Niethammer (for the authors.)

REVIEWER COMMENTS

Reviewer #1 (Remarks to the Author):

In this article, Gelashvili and colleagues investigate the mechanisms by which wounds are detected and responses are induced, using Zebrafish as a model system. They use fluorescent dextran in the blood circulation to measure leakage from the blood vessels and from the wound. Careful quantification and use of the right controls shows that leakage from the blood vessels is induced by the osmotic shock accompanying the wound and requires Alox5a, suggesting a role for the cPLA2 pathway. The authors then show that immune cells, more specifically macrophages, are needed to induce leakage of dextran from blood vessels upon wounding, and that cPLA2 is translocated to the nuclear envelope in macrophages rapidly after a wound; they also show that cPLA2 mutant animals are defective in the induction of leakage by wounding. They propose a working model by which osmotic stress induced by the wound rapidly induces swelling of cells and nuclei in the tissue, leading to cPLA2 activation in macrophages, production of arachidonic acid by these cells, which promotes blood vessel leakage via Alox5a. Overall the article is very clear, contains carefully executed experiments and controls, and provides a very nice demonstration of how the activation of cPLA2 in immune cells contributes to the response of tissues to wounds. I would thus recommend publication of the article. I have just a few minor concerns/questions.

1) The authors write that they observe recruitment of cPLA2 to the INM of macrophages, referring to figure 4. While I understand that other articles demonstrate well this recruitment, it is not clear in figure 4. Maybe the authors mean enrichment at the nuclear periphery? This is quite clear from the quantifications, but it is hard to see how the INM can be distinguished here.

Response: Thank you for pointing this out. We now use contrast-adjusted single-plane confocal slice views instead of MIPs for the insets of Fig. 4c, 4e which makes the rim localization of cPLA₂ easier to discern in the still images. Also please refer to Fig.S5g for a zoomed-in view on cPLA₂ translocation. Furthermore, the profile plots (derived from the respective maximal intensity projection images), as well as the corresponding supplementary movies 8, 9 further highlight the translocation events quantitatively and dynamically.

2) In most perturbations, the amplitude of the reduction in dextran leakage appears quite small. The authors claim that the pathway they identified is 'required' to induce this leakage. Maybe they could rather say that it 'contributes'. It is not clear for me (I am not a specialist of this phenomenon nor these techniques), whether the reduction they observe would correspond to what is expected in the case of a total lack of leakage induction, or it is really just a partial effect, meaning that the cPLA2 pathway in macrophages is just one of the signals inducing this leakage. This should be clarified in the discussion of the results (but it could also be that I misunderstood something here).

Response: We agree and adjusted the presentation and discussion of our results for better judgement of effect size.

In the former version, our vessel leakage (lv) plots only depicted the *overall leakage without baseline correction, that did not easily allow to assess wound-induced-leakage, which is of main interest*. Given the way we calculate lv(tot/norm) as area under the curve of our normalized lv plots, there is always a 3600 AU theoretical baseline included (now indicated by a dotted line as thoroughly explained in the revised results section). Only values above 3600 denote wound induced leakage, everything below is baseline leakage (which we also confirmed by the inclusion of unwounded animals). Using this baseline, osmotic inhibition of vessel permeabilization is ~90%, and all other effects around 60%. While these effects are still partial, they are nevertheless substantial.

We hope that our revised representation and discussion better highlights the importance of physically stimulated eicosanoids in rapid vessel permeabilization and the key role of macrophages in transducing these signals. We adjusted the discussion accordingly, and again apologize for the confusion.

3) The authors showed several elements of their working model independently of each other. If I understood correctly, they showed that macrophages contribute to the response, that cPLA2 also contributes (although the depletion is in the whole animal and not specifically cPLA2) and that they observe cPLA2 enrichment at the nuclear periphery in macrophages (but they did not seem to have stained other cell types). So they never test directly a role of cPLA2 in macrophages, and they also do not exclude a contribution of cPLA2 in other cells (like endothelial cells) - is cPLA2 also recruited to the nuclear periphery in endothelial cells, or in other cells of the tissue? Can they deplete cPLA2 in specific cell types? If more specific

experiments are not feasible, I still think that their working model is not contradicted by their experiments, but they might want to tone down their conclusions and make it clear that there are other possible contributions to the induction of leakage upon wounding (like cPLA2 activation directly in other cell types, or indirectly, and the fact that there is no direct proof of the contribution of cPLA2 specifically in macrophages). Again, I apologize in case of misunderstanding on my side - this system is quite complex and I might have missed important elements of proof for the working model proposed.

Response: Thank you for your thoughtful comments. As suggested, we now studied cPla2 translocation in endothelial cells and found no significant nuclear membrane translocation above background (Fig. S7a-c) showing that macrophages are more sensitive to osmotic signals than other cell types at similar distances from the wound.

We attempted macrophage transplantation experiments (as suggested by rev #2) to test whether introducing wt-macrophages into cPla2 pathway mutants would rescue the leakage defects. For such experiment to yield informative results, we need to be able to transfer a sufficient number of macrophages and ensure that they reach the perivascular tail fin region, which turned out to be very difficult in our hands.

We were able to isolate $\sim 5 \times 10^5$ viable fluorescent macrophages from >200 fish. However, this number was still far below the $\sim 10^7$ cells typically required for robust transplantation experiments. The recipient embryos developed severe edema, enlarged hearts, and swollen anterior vessels overnight, and we could not observe any of the transplanted macrophages in the perivascular space. Despite of our intense efforts, we were not able to turn larval macrophage transplantation into a robust and reproducible approach that we trust to report here.

We cannot exclude a cPla₂ activation in other cells than macrophages may also contribute to arachidonic acid production and cPLA2-ALOX5-LTA4H dependent vessel permeabilization. In fact, we previously showed that cPla₂ at the zebrafish wound margin, also by physical signals, just as in macrophages shown here ^{2,3}. Thus, arachidonic acid could, in principle, diffuse from the wound margin to the perivascular macrophages to at least partially fuel their eicosanoid synthesis. But given that (i) macrophages are well-known to contain the full cPLA2-ALOX5-LTA4H pathway, (ii) activate cPla2 within seconds of osmotic shock, and (iii) are required for the sensitivity of vessel permeabilization to ALOX5 inhibitors, the balance of evidence

suggests that macrophages are the most likely cells converting the physical wound signal (osmotic shock) into a vessel permeabilizing lipid. This conclusion does not exclude the possibility that other cell types contribute to this process. We tried to clarify this notion with a new overview scheme and balanced discussion.

Reviewer #2 (Remarks to the Author):

In this work, Gelashvili et al., presents evidence that macrophages localizing to the perivascular space of embryonic zebrafish blood vessels trigger a quick response to wounding by regulating osmotic signals and vascular permeability. The work addresses an interesting and important aspect of wound healing and introduces a novel signaling pathways into the mechanism of tissue injury. The article is written in clear and detailed language, the data is of good quality and quantifications were carefully performed, however, there are several conclusions that would require further validation to support the author's conclusions more directly, for example, a characterization describing the vascular phenotypes of the mutants described in this study under homeostasis would be a key point to interpret the results presented by the authors, as this would provide insights into the mutants vascular integrity, architecture and function. Some of my concerns and questions are described below:

Figure 1

- Is there also response to the wounds caused by intracardial injection?

Response: This is a great question. We now analyzed vessel leakage in 70kDa dextran injected animals without tail fin wounding. Injected but non-wounded animals at HYPO show almost the same, or slightly less vessel leakage as wounded ISO, or ISO+ animals (Fig. 1d). These data suggest only minimal if any vessel leakage caused by the injection procedure.

- Dextran tend to diffuse to tissues outside the vasculature – which is evident in all figures – the amount of diffusion may vary from fish to fish based on artifacts like the injury induced at the injection site, or the minimal differences in tail clipping sizes, the authors should show non-wounded animals as controls and include the values obtained in their quantifications. Alternatively, they could use fluorescent qdots or nanobeads which stay in circulation for longer periods of time without diffusing to adjacent tissues and have

been extensively used to label vascular structures in zebrafish. Any diffusion would be strictly at the wound site which could facilitate a more restricted quantification, but also help visualizing vascular phenotypes much more clearly.

Response: We agree and now included slow-diffusing 500 kDa dextran (instead of Q-dots) and non-wounded animals into the analysis as requested (Fig. S1j, S1l, S2g, S2h, S5f). Comparable to Q-dots, and unlike 70 kDa dextran, 500 kDa dextran remains nicely contained within the vasculature highlighting its morphology. In injured animals, 500 kDa dextran leaks from the vessel regions nearby the amputation site (Fig. S1j-m), which we now quantified either by leakage intensity or thresholded leakage area at two different objective magnifications (20x: Fig. 1j-k; 63x: l-m). As with 70 kDa dextran, the leakage is largely abrogated by ISO bathing. Furthermore, our 500 kDa experiments in unwounded mutants (20x: Fig. S2g-h, S5f) did not indicate obvious differences in baseline vessel morphology or permeability.

- It would be helpful to relabel this figure, assigning each panel a letter for easier reading, although this version is properly described, having single letters for each panel will make the manuscript easier to follow especially in figures with multiple panels.

Response: In the previous version the logic of our presentation was indeed a bit flawed, and we apologize for the confusion this may have caused. Namely, the representative images were not grouped with their respective quantification, which is not ideal, and might have been annoying. We revised all our main figures now for better readability, and less subpanel jumping.

Figure 2

- What is the vascular phenotype in the *alox5a*, *alox12* and *ita4h* mutants? The few vessels showing in this figures hint of vessels having significant architectural differences (figure 2b,c and d), where they appear much thinner and less branched or developed, potentially explaining why some of these mutants (*alox12* and *Ita4h*) showed reduced vessel leakage.

Response: Vessel morphology is difficult to reliably assess using 70 kDa dextran, given its high baseline tissue leakage and fast diffusion. To better evaluate vascular baseline phenotypes, we thus now imaged 500 kDa

dextran in non-wounded animals and observed no gross differences in vessel morphology and baseline permeability comparing alox5a, ita4h, or pla2g4aa mutants with their wild-type siblings (see 2nd response to Fig. 1 comment).

Is the vasculature in these mutants leaky in general?

Response: Our new experiments indicate that mutants and wt siblings show no significant baseline differences in leakage (see above).

Once again, showing an uninjured embryo's vasculature labeled with qdots could provide insights into the vascular phenotypes of these mutants. This is also applicable to all the drugs quantified in Figure 2, where no embryo data is shown to support their vascular leakage observations ->

Response: We completely agree and have addressed this concern. Our 500 kDa dextran experiments do not indicate gross differences in vascular morphology or permeability in unwounded animals (see above). For the pharmacological treatments and vessel leakage, representative larva images are now provided for 70 kDa dextran for antagonist (Fig. 2h; Fig. S3a) and agonist (Fig. S3f, g). Licofelone-treatment had no gross effect on vessel morphology as judged by visual inspection of KDRL-eGFP images or calculation of the vessel/tail fin area ratio (Fig. S3d, S3e).

Figure 3

- Based on references to zebrafish transcriptomics the author mention that alox5a and ita4h are expressed by immune cells – potentially macrophages and neutrophils, however they do not show a staining validating this, which makes their ablations informative but not conclusive on the specific role of these genes in vessel permeabilization –

Response: This is correct. Unfortunately, there are no specific antibodies for these zebrafish proteins commercially available that would allow to conduct reliable immunofluorescence or WB staining. Establishing new, reliable Abs for zebrafish proteins is generally desirable, but would constitute a considerable financial resources and time that are unfortunately beyond the scope of this study. However, mutually consistent zebrafish in situ hybridization and mRNAseq data supporting the leukocyte localization of these enzymes in zebrafish are already available from multiple independent sources.

We are not sure that simply repeating those experiments will provide more insights.

Based on all the available literature from mammals and zebrafish, the cPLA2-ALOX5-LTA4H is indeed highly conserved macrophages from fish to humans. In mammals, the biochemical activity of the cPLA2-ALOX5-LTA4H eicosanoid branch is typically assessed in isolated (e.g., peritoneal) macrophages. We now provide a more thorough discussion and literature account of this work.

as many other factors coming from these immune cells could lead to their results. Similarly, expression of these genes by non-immune cells was not tested.

Response: Our data show that (i) rapid vessel permeabilization is almost completely (~90%) mediated through osmotic cues (Fig. 1d), (ii) largely (~60%) mediated by the cPLA2-ALOX5-LTA4H pathway and macrophages (Fig. 2, 4; S7). They further show that (iii) macrophages are required for vessel regulation through this bioactive lipid pathway, and that (iv) the same pathway is directly activated by osmotic shock in macrophages. We think this balance of evidence argues that macrophages are main players in converting physical wound signals into rapid vessel responses. Nevertheless –as the reviewer noted—we cannot exclude that other cells and pathways may contribute, too. Namely, ~40% of the vessel response is currently unaccounted by our genetic perturbation. We only know that neutrophils do not contribute to these unaccounted events. We revised our discussion and overview scheme (Fig. S7d) to better clarify these notions.

- As the authors have already made alox5a and Itah4 mutants, they could repeat these experiments on these animals or test their hypothesis transplanting mutant macrophages/neutrophils into wild type animals and testing whether these mutant cells affect vascular permeability/leakage.

Response: We agree that these would be great experiments and thus attempted to establish the outline transplantation approach. By FACS sorting, we were able to isolate $\sim 5 \times 10^5$ viable fluorescent macrophages from >200 fish. Unfortunately, this number was still below the $\sim 10^7$ cells typically required for robust transplantation experiments. The recipient embryos developed severe edema, enlarged hearts, and swollen anterior vessels

overnight, and we could not observe any of the transplanted macrophages in the perivascular space. Regrettably, and despite of our intense efforts, we were not able to turn larval macrophage transplantation into a robust and reproducible approach that we trust to report here.

- Here the authors mentions that the NTR/MTZ dependent ablation of neutrophils did not modulate vessel permeability, it would be good to see examples of these treated animals and the general distribution of their immune cells comparing inside and outside the vasculature. –

Response: Thank you for the suggestion. To address the distribution immune cells of MTZ-treated animals, we provide examples of NTR2.0-lyz:MTZ ablation of neutrophils together with Sudan black staining post-wounding. Using our described protocol, neutrophils were eliminated within 24 h, as demonstrated by the absence of Sudan black–positive cells in every MTZ-treated embryo (Fig. S4l, S4m).

- It is also not clear how far are macrophages and neutrophils migrating in response to the wound and the change in osmotic solutions?

Response: We now quantified the requested leukocyte migration patterns. (Fig. S4g, S4h). Based on the literature and these results, osmotic shock mobilizes leukocyte that are initially up to ~200 μm away from the wound⁸⁻¹¹. In terms of directionality, neutrophils show a strong bias toward the wound, whereas macrophages display broader directional variability.

Figure 4

- Once again, showing the vascular phenotype of the cPla2 mutants would be helpful to get a sense of the vascular architecture and integrity.

Response: We now show the vascular phenotypes in age-matched siblings for both 70kDa (Fig. 4a) and 500kDa (Fig. S5f). We did not note reproducible differences.

- The author mention ‘perivascular macrophages’ during their nuclear shape sensing analysis in cPla2 transgenics, and elaborate on this in the discussion, it would be helpful to characterize these cells and determine whether these are classic mpeg1+ macrophages localized outside vessels

or actual perivascular immune cells with a specific cellular profile (e.g. mrc1/lyve1+)? -> both of these are membrane glycoproteins –

Response: We agree. What determines the increased osmotic sensitivity of these mpeg1+ cells nearby vessels is one of the most intriguing questions raised by our current work. The mpeg1 promoter we use here is perhaps the most well established and preferred macrophage reporter in the zebrafish field, so the 0-hypothesis is that these “perivascular” macrophages are just embryonic macrophages/monocytes neighboring blood vessels. Any other statement, e.g., the interesting reviewer hypothesis that these could be specialized perivascular macrophages, would require an extraordinary amount of evidence, likely comprising an entirely new project.

At this time, we are not aware of any zebrafish antibodies (or cell-specific promoters) that would allow us to selectively stain and sort the perivascular mpeg1+ population for further mRNA profiling. This could answer the reviewer’s question. Clearly, developing better tools to enable such approach (e.g., photoconversion and sorting of perivascular cells using robotic high throughput microscopy paired with machine learning) is a consequent next step transcending the scope of this focused osmotic vessel permeabilization study. We have revised our discussion to clarify that our usage of “perivascular” does not allude to a specific differentiation state, although this possibility is not excluded and exciting.

- A general comment is that many of these experiments were performed on the casper mutants, do these mutants have any alterations in the way wound healing occurs, or immune cells react to wounding? the authors should either show this mutant background does not affect their results (compared to a WT) or reference studies pointing out to previous reports where this was addressed.

Response: To experimentally address this comment, we extended our analyses from the casper background to AB animals and now provide new data with 70kDa dextran (Fig. S1g–i) showing that the casper and AB backgrounds respond similar with respect to wound-induced vessel leakage.

Reviewer #3 (Remarks to the Author):

The manuscript Gelashvili, et al, focuses on understanding how macrophages signal to respond to wound signals and initiate rapid vascular repair. Though the data is interesting, new insights since previously published work seem limited for the scope of Nature Communications.

Response: We are saddened by the reviewer's discouraging overall assessment of our work, though we admit that we could have done a much better job of highlighting its relevance in the first version. Thus, we have substantially revised our manuscript based on rev #1 & 2's constructive critiques.

We respectfully disagree with this reviewer's significance and novelty assessment. Neither we nor others have previously reported a role for macrophages in osmotic vessel permeabilization and the cPLA2-ALOX5-LTA4H pathway, and nuclear membrane mechanotransduction *in vivo*.

The role of the nucleus as a mechanosensor and the contribution of nuclear mechanotransduction to physiological processes is still a largely uncharted field mostly studied in reconstituted cell culture environments, not in live animals – as we do here.

To clarify the broad relevance of our findings, we have completely revised our result and discussion sections.

The imaging is carried out at fairly low resolution

Response: We now include also a higher magnification experiment, alongside our standard 20x mag approach. Notably, 20x mag imaging allows to image larger tissue fields at higher temporal resolution than possible with 63x (Fig. S1j-m).

and while the responses to changes in osmotic stress are pertinent, the phenotypes are modest and definitive links to mechanics and the proposed genetic pathway are lacking.

Response: We admit that the previous presentation of our leakage (Iv) data might have caused confusion, as we did not explicitly distinguish baseline from wound-induced leakage. Indeed, this was a severe shortcoming that we now have addressed by indicating the leakage baseline in all applicable graphs. We now also experimentally determined baseline leakage in non-wounded animals. We estimate that wound-induced vessel permeabilization is (i) almost completely (~90%) caused by osmotic cues, (ii) and to ~ 60%

via the cPLA2-ALOX5-LTA4H pathway and macrophages. We would not consider ~60% inhibition modest, although it is indeed partial. We have balanced our discussion to highlight this notion.

We regret that we might have contributed to this harsh reviewer judgement by not more explicitly discussing effect sizes in our earlier version, and due to unclear data presentation.

The paper is overly technical to read, and somewhat hard to follow the logic of. I appreciate the immense amount of work presented; however, in this format, the paper is not well targeted for general readership.

Response: We completely revised our results and discussion to reduce the use of technical language. The text should be now much more readable.

References:

1. Jeong, J.-Y. *et al.* Functional and developmental analysis of the blood–brain barrier in zebrafish. *Brain Research Bulletin* **75**, 619-628 (2008).
2. Enyedi, B., Kala, S., Nikolich-Zugich, T. & Niethammer, P. Tissue damage detection by osmotic surveillance. *Nat Cell Biol* **15**, 1123-1130 (2013).
3. Enyedi, B., Jelcic, M. & Niethammer, P. The Cell Nucleus Serves as a Mechanotransducer of Tissue Damage-Induced Inflammation. *Cell* **165**, 1160-1170 (2016).
4. Wattrus, S.J. & Zon, L.I. Stem cell safe harbor: the hematopoietic stem cell niche in zebrafish. *Blood Adv* **2**, 3063-3069 (2018).
5. Speir, M.L. *et al.* UCSC Cell Browser: visualize your single-cell data. *Bioinformatics* **37**, 4578-4580 (2021).
6. Sur, A. *et al.* Single-cell analysis of shared signatures and transcriptional diversity during zebrafish development. *Dev Cell* **58**, 3028-3047.e3012 (2023).
7. Wang, R. *et al.* Construction of a cross-species cell landscape at single-cell level. *Nucleic Acids Research* **51**, 501-516 (2023).
8. Niethammer, P., Grabher, C., Look, A.T. & Mitchison, T.J. A tissue-scale gradient of hydrogen peroxide mediates rapid wound detection in zebrafish. *Nature* **459**, 996-999 (2009).
9. Katicaneni, A. *et al.* Lipid peroxidation regulates long-range wound detection through 5-lipoxygenase in zebrafish. *Nat Cell Biol* **22**, 1049-1055 (2020).
10. Walters, K.B., Green, J.M., Surfus, J.C., Yoo, S.K. & Huttenlocher, A. Live imaging of neutrophil motility in a zebrafish model of WHIM syndrome. *Blood* **116**, 2803-2811 (2010).
11. Li, L., Yan, B., Shi, Y.Q., Zhang, W.Q. & Wen, Z.L. Live imaging reveals differing roles of macrophages and neutrophils during zebrafish tail fin regeneration. *J Biol Chem* **287**, 25353-25360 (2012).

12. Shiau, C.E., Kaufman, Z., Meireles, A.M. & Talbot, W.S. Differential Requirement for irf8 in Formation of Embryonic and Adult Macrophages in Zebrafish. *PLOS ONE* **10**, e0117513 (2015).
13. Ferrero, G. *et al.* The macrophage-expressed gene (mpeg) 1 identifies a subpopulation of B cells in the adult zebrafish. *J Leukoc Biol* **107**, 431-443 (2020).
14. Ellett, F., Pase, L., Hayman, J.W., Andrianopoulos, A. & Lieschke, G.J. mpeg1 promoter transgenes direct macrophage-lineage expression in zebrafish. *Blood* **117**, e49-56 (2011).

We would like to thank the reviewers for their interest and thorough assessment of our manuscript.

Reviewer #1 (Remarks to the Author):

In their revised article, the authors have provided new data and analysis, including more controls and clarification of level of wound induced leakage, also being more subtle in their claims to leave open some alternatives when they did not provide perfect direct proofs - for example for the fact that the downstream products of the cPLA2 pathway are mostly coming from perivascular macrophages. They tried some direct transplantation experiments that failed, and as I am not a zebrafish expert, I unfortunately have no specific idea to suggest to go around this experiment to test their hypothesis more directly. I think that, despite this failure for a direct proof, their article provides very interesting and convincing data to support their working model, and that the way they now present their results and discuss them, is well adapted to the level of confidence they can have in their results. I would thus recommend publication without further revisions.

> Thank you very much for lending your scientific perspective as it helped us strengthen and improve the manuscript!

Reviewer #3 (Remarks to the Author):

The included revisions have decreased my initial concerns, and dramatically improved the manuscript

> Thank you!

Reviewer #4 (Remarks to the Author):

I have been appointed to replace Reviewer #2, who are no longer available assess the authors' response to their previous comments.

After reviewing the revisions that the authors performed, I believe that the authors have in general satisfactorily addressed the Reviewer #2 comments.

However, I would like to raise a few points:

1. Given that the images of blood vessels are low in magnification and resolution, it is challenging to determine whether there is a vascular phenotype

in the *alox5a*, *alox12* and *ita4h* mutants; and between controls and after osmotic shock. Based on this resolution, there is no observed gross differences at the tissue scale, as the authors claim, although the ventral caudal vein of *alox5a* mutant appears more dilated than the wildtype embryo to me (Supplementary Fig. 2).

> Thank you for the note! We think we have quantitatively addressed this point in Fig. S1j-k and Fig. S6a-c

We agree. The Fig. S1j was full field of view, which we now cropped and enlarged for better visibility. To clarify this point, we now include a new supplemental figure with enlarged, native resolution (non-rasterized or kept at 300ppi) insets (showing vessel morphology) for better visibility. From these enlarged, higher resolution images it should be clear that there are no overt morphological differences in baseline vessel morphology between wild types and mutants.

In the new Fig. S6, we enlarged the quantitative panels showing that there is also no difference in basal vessel area ($I_{v_{area}}$, = morphological proxy) and vessel permeability ($I_{v_{tot/norm}}$, functional proxy). The morphological differences mentioned by the reviewer are most likely within the range of normal biological variability.

2. I would have liked to see the effects of osmotic shock (after injury) on endothelial cell-cell junctions (morphology) and upon the loss of macrophages and/or perturbation of the cPLA2-ALOX5-LTA4H pathway. This will provide some insights into how blood vessel permeability is increased.

> Indeed, the next big question is how the blood vessel permeability is achieved, e.g., through chemical or physical interactions of the perivascular macrophages with the blood vessels.

Currently, we do not have appropriate genetic marker tools to visualize endothelial–interstitial leakage routes in vivo at sufficient spatiotemporal resolution to make solid mechanistic statements. The leakage possibly involves junctional remodeling or trans-endothelial transport (e.g., caveolae). Carefully testing these ideas, e.g., by high-speed/high-resolution intravital microscopy would be a new, dedicated project that is currently beyond our technical capabilities.

3. The authors stressed the function of nuclear membrane mechanotransduction in transmitting wound signals to blood vessels and emphasized the importance of inner nuclear membrane (INM) tension in recruiting cPLA2 to the INM after osmotic shock. The evidence supporting this claim is rather weak in this manuscript. This is partly because it is not possible to see the shape of cell nuclei as well as the shape of the macrophages, thus making it difficult for non-macrophage experts to determine the subcellular localization of cPla2-mKate2. I would have appreciated a more detailed examination of cPla2 localization with, for example, a nuclear/nuclear membrane marker to clearly show that cPla2 redistributed from the nucleoplasm to the nuclear membrane. Along this line, why does cPla2-mKate2 distribution look different in Fig. 4e (more elongated, like the cell body) compared to Fig. 4c (more circular)? Does HYPO condition change the distribution of cPla2 or the shape of the nucleus?

>Thank you for the comment. To clarify: The constitutive, nuclear localization of zebrafish cPla2 is well established (Enyedi et al., NCB, 2013; Enyedi et al., Cell, 2016). In addition, we provide with Fig. S5f a magnified representative image of a perivascular macrophage with the ER marker (KDEL-eGFP, yellow). Since the ER and nuclear membrane are contiguous, this marker outlines the nucleoplasm. Before the translocation (left column), one can clearly see that cPla2 (magenta) is within the nucleoplasm, before it reversibly translocates to the nuclear membrane (middle column) as marked by KDEL-eGFP (yellow). The same translocation (please note the change from homogenous fluorescence distribution to rim-like fluorescence distribution) is also visible in the magnified insets of Fig. 4c and S7a,c), and quantified in the respective profile plots (Fig. 4d; S7b-d). We have now added additional annotations to these rather complex figures to better guide the reader's attention and avoid misunderstandings.

Hypotonic shock swells the nucleus (see Enyedi et al., Cell, 2016), but this momentous volume increase upon shock is difficult to perceive or measure at our spatiotemporal resolution. The macrophage nuclei have complex, highly variable shapes depending on how they are confined by the surrounding tissue at a given moment. The noted shape differences between 4c (hypo, round), and 4e (iso, elongated) most likely reflect biological variability. This notion is corroborated by Fig. S5f (hypo, elongated) and S7 (hypo, more elongated; iso, more rounded), which show opposite shape trends than Fig.4c/e.